# Interaction of 7SK with the Smn complex modulates snRNP production

Changhe Ji[1], Jakob Bader [2], Pradhipa Ramanathan[3], Luisa Hennlein[1], Felix Meissner [2,4,5], Sibylle Jablonka [1], Matthias Mann [2,6], Utz Fischer [3], Michael Sendtner [1,7✉] & Michael Briese [1,7✉]

Gene expression requires tight coordination of the molecular machineries that mediate transcription and splicing. While the interplay between transcription kinetics and spliceosome fidelity has been investigated before, less is known about mechanisms regulating the assembly of the spliceosomal machinery in response to transcription changes. Here, we report an association of the Smn complex, which mediates spliceosomal snRNP biogenesis, with the 7SK complex involved in transcriptional regulation. We found that Smn interacts with the 7SK core components Larp7 and Mepce and specifically associates with 7SK sub-complexes containing hnRNP R. The association between Smn and 7SK complexes is enhanced upon transcriptional inhibition leading to reduced production of snRNPs. Taken together, our findings reveal a functional association of Smn and 7SK complexes that is governed by global changes in transcription. Thus, in addition to its canonical nuclear role in transcriptional regulation, 7SK has cytosolic functions in fine-tuning spliceosome production according to transcriptional demand.

[1] Institute of Clinical Neurobiology, University Hospital Wuerzburg, Wuerzburg, Germany. [2] Department of Proteomics and Signal Transduction, Max Planck Institute of Biochemistry, Martinsried, Germany. [3] Department of Biochemistry, Theodor Boveri Institute, University of Wuerzburg, Wuerzburg, Germany. [4] Experimental Systems Immunology, Max Planck Institute of Biochemistry, Martinsried, Germany. [5] Department for Systems Immunology & Proteomics, Institute of Innate Immunity, University Hospitals, University of Bonn, Bonn, Germany. [6] NNF Center for Protein Research, Faculty of Health Sciences, University of Copenhagen, Copenhagen, Denmark. [7] These authors jointly supervised this work: Michael Sendtner, Michael Briese. ✉email: Sendtner_M@ukw.de; Briese_M@ukw.de

Splicing of pre-mRNAs fulfills an import function in all higher organisms by increasing the complexity of gene products. In eukaryotic cells, splicing of pre-mRNAs is tightly coupled to transcription. Nascent pre-mRNAs are processed co-transcriptionally by the spliceosome which removes introns and joins respective exonic sequences. The processing of individual pre-mRNAs in the nucleus is highly coordinated with their transcription, such that the kinetics of transcription can influence splicing fidelity and vice versa[1]. In mammalian cells, exon definition mechanisms facilitate precise removal of introns and involve the activity of RNA polymerase II which aids in co-transcriptional recruitment of the spliceosome through its C-terminal domain[2]. While such direct links between transcription and spliceosomal processivity have been investigated in detail, it has remained less clear whether additional mechanisms exist whereby the synthesis of the spliceosomal machinery itself is adjusted according to the global transcriptional demand of a cell.

The spliceosome is composed of small nuclear ribonucleoproteins (snRNPs) each of which consists of a snRNA, a set of seven Sm proteins common to all snRNPs (except U6) and several snRNP-specific proteins[3]. The biogenesis of snRNPs occurs in the cytosol and requires the Survival Motor Neuron (SMN) complex[4,5]. This complex is composed of SMN and GEMIN2-8[6]. Following transcription in the nucleus, snRNAs are exported to the cytosol where they associate with the SMN complex. The Sm proteins are preassembled by pICln and then transferred by the SMN complex onto the snRNAs, forming a heptameric ring around a uridine-rich sequence[7]. Subsequently, the 5′ cap of snRNAs is hypermethylated for re-import of the snRNPs into the nucleus[8].

SMN deficiency due to deletions or mutations of the *SMN1* gene causes spinal muscular atrophy (SMA)[9], a neurodegenerative disease that primarily affects spinal motoneurons. Even though the functions of the SMN complex in snRNP biogenesis have been studied in great detail[10] much less is known about how the activity of the SMN complex is regulated. A previous study showed that in mice the activity of the Smn complex in snRNP assembly is particularly high in brain and spinal cord during development and decreases between 2–3 weeks of age in these tissues[11]. Likewise, SMN complex activity is high in human neuronal precursor cells but decreases during neuronal differentiation[11]. Thus, SMN complex activity is high when transcription is high such as in proliferating cells and downregulated in postmitotic adult cells with lower transcriptional activity. However, the underlying molecular mechanisms linking cytosolic SMN complex activity to transcription have remained obscure.

The transcription of pre-mRNAs through RNA Polymerase II is controlled by 7SK, a highly structured, abundant non-coding RNA of 331 nucleotide length[12]. 7SK is stabilized by methyl phosphate capping enzyme (MePCE) at the 5′ end and La-related protein 7 (LARP7) at the 3′ end[13,14]. MePCE adds a methyl group to the γ-phosphate of the 5′ guanosine of 7SK, and remains bound to stem-loop (SL) 1 of 7SK, thereby stabilizing its 5′ end[14–16]. LARP7 binds to SL4 of 7SK and protects its 3′ end from exonucleolytic degradation[15,17,18]. Together, 7SK, MePCE, and LARP7 form a "core" RNP platform to which additional proteins can bind in a reversible manner. In the nucleus, 7SK regulates transcriptional elongation by sequestering the positive transcription elongation factor b (P-TEFb)[19,20]. For a large subset of genes, RNA polymerase II tends to pause downstream of the transcription initiation site shortly after transcription initiation[12]. P-TEFb, a heterodimer of CDK9 and Cyclin T1, releases paused Polymerase II by phosphorylating its C-terminal domain as well as by phosphorylating the transcriptional repressors DRB sensitivity inducing factor (DSIF) and negative elongation factor (NELF)[21]. Binding of 7SK to P-TEFb inhibits this activity. This function requires HEXIM1 which associates with SL1 of 7SK[22,23].

More recently, a number of RNA-binding proteins have been identified as 7SK interactors by proteomics approaches including the heterogeneous nuclear ribonucleoproteins (hnRNPs) A1, A2/B1, Q and R, and RNA helicase A (RHA)[24,25]. In agreement, we identified 7SK as the top interacting RNA of hnRNP R by individual nucleotide resolution crosslinking and immunoprecipitation (iCLIP)[26]. The 7SK/hnRNP complexes are separate from 7SK/HEXIM1/P-TEFb complexes and the balance between these 7SK subcomplexes is determined by the transcriptional activity of a cell[24]. Upon transcriptional inhibition, lack of nascent RNA promotes binding of hnRNPs to 7SK, thereby dissociating HEXIM1 and P-TEFb from 7SK[27]. Vice versa, increased cellular transcription promotes binding of hnRNPs to newly synthesized RNAs such that P-TEFb can be sequestered by 7SK[27].

In this study, we report that 7SK and hnRNP R regulate Smn complex activity, thus providing a link between the regulatory role of 7SK in transcription and the function of Smn in snRNP biogenesis. We found that the Smn complex interacts with 7SK RNPs in NSC-34 motoneuron-like cells and primary embryonic mouse spinal motoneurons. Importantly, Smn selectively associated with 7SK/hnRNP but not with 7SK/Hexim1/P-TEFb complexes. Transcriptional inhibition enhanced the association of Smn with 7SK particles which then led to reduced cellular snRNP levels. The interaction of Smn with 7SK RNPs containing hnRNP R occurred in the cytosol, supporting a role of this 7SK complex in regulating the snRNP assembly functions of Smn. Together, our results indicate that 7SK particles containing Larp7, Mepce, and hnRNP R regulate Smn complex activity, thus providing a mechanism how transcriptional control is linked to the biogenesis of the spliceosomal machinery.

## Results

**Smn interacts with 7SK/hnRNP complexes.** Based on previous observations that 7SK/hnRNP R complexes are present in cytoplasmic fractions of NSC-34 cells and motoneurons[26], we searched for new protein interactors of this non-coding RNA in NSC-34 cells, a cell line with motoneuron-like properties[28]. For this purpose we used a biotinylated RNA antisense oligonucleotide to pull down endogenous 7SK complexes from cell lysates. In order to enrich for interactors of the 7SK complex at functional states when the complex does not regulate P-TEFb activity, we used an oligonucleotide (Biotin-pd7SK) that targets nucleotides 17–33 within SL1 of mouse 7SK, corresponding to the region which Hexim1 normally binds to as part of 7SK/Hexim1/P-TEFb complexes[23,24]. Consequently, Biotin-pd7SK preferentially targets 7SK complexes not containing Hexim1 and P-TEFb and, therefore, can be used to selectively pull down 7SK/hnRNP complexes[24]. In support of this notion, we found that, in addition to Larp7 as part of the 7SK core RNP, Biotin-pd7SK co-purified hnRNP R and hnRNP A1 but not Hexim1 (Supplementary Fig. 1a). As control we used a scrambled oligonucleotide (Biotin-Scr) which did not purify the 7SK complex indicated by lack of Larp7 co-purification. We also investigated the interaction of Biotin-pd7SK with other abundant coding and non-coding RNAs by quantitative PCR (qPCR) of the co-precipitate. As a result, we observed efficient precipitation of 7SK but not Malat1, β-actin mRNA (*Actb*), U1 snRNA, 7SL, or *Gapdh* (Supplementary Fig. 1b).

In order to identify novel 7SK interactors we performed proteomic analysis of proteins co-precipitated by Biotin-pd7SK and Biotin-scr from NSC-34 lysate (Supplementary Fig. 1c and Supplementary Data 1). Among proteins that were enriched by Biotin-pd7SK relative to Biotin-scr were Larp7 and Mepce (Fig. 1a and Supplementary Data 2), indicating that Biotin-pd7SK efficiently purified 7SK-associated proteins. Additionally, we detected hnRNP A1, A2/B1, Q, and R, but not RHA, in our

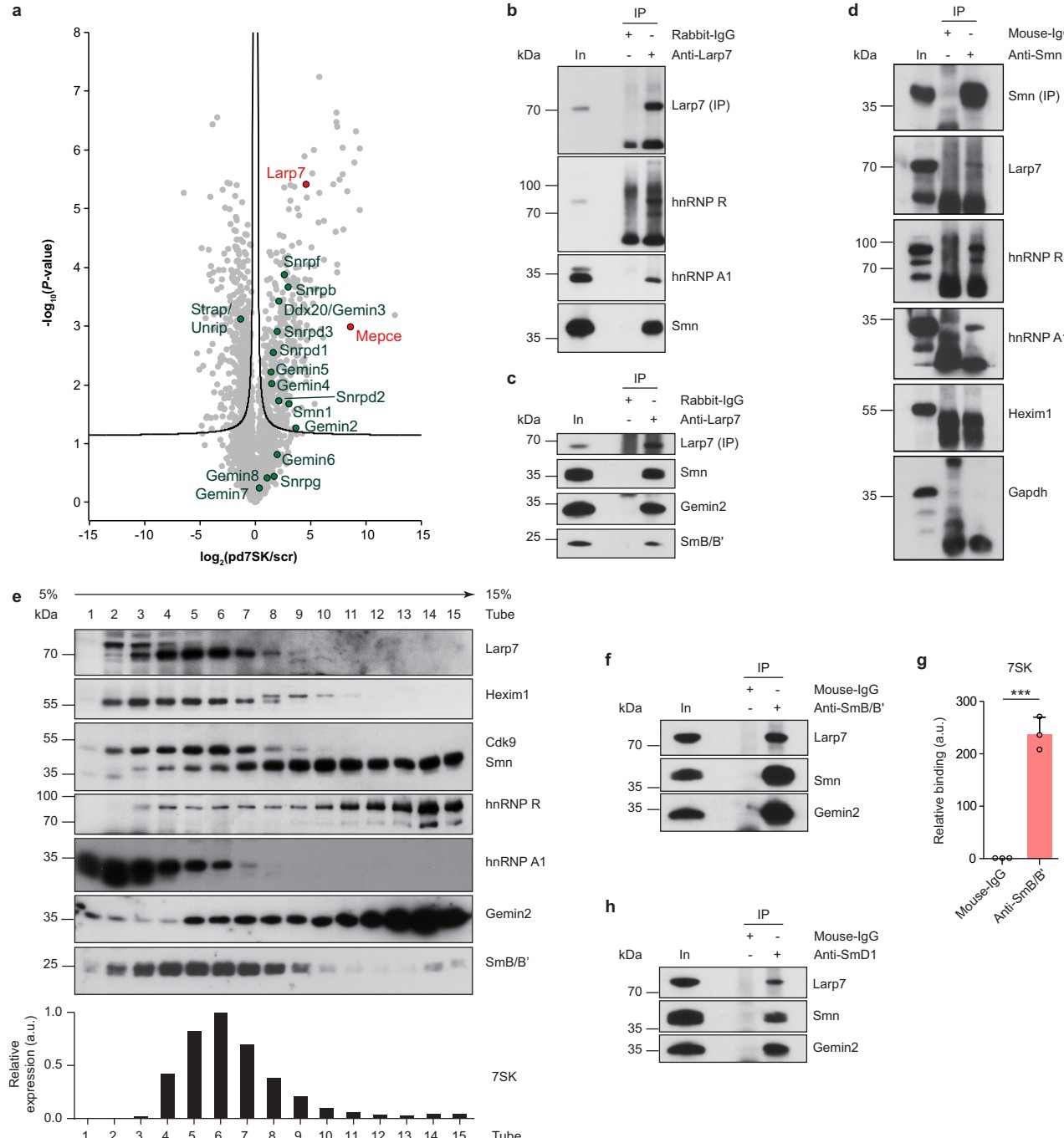

**Fig. 1 Smn interacts with 7SK/hnRNP but not with 7SK/P-TEFb complexes. a** Volcano plot of the 7SK interaction proteome following pulldown with Biotin-pd7SK and scr control from NSC-34 cell lysate. Smn complex proteins are highlighted in green, the 7SK core interactors Mepce and Larp7 are marked in red. **b** Western blot analysis of proteins co-immunoprecipitated from NSC-34 cells by an anti-Larp7 antibody directed against the C-terminus of Larp7. Immunoprecipitation with rabbit-IgG antibody was used as control. In, input; IP, immunoprecipitation. **c** Co-immunoprecipitation of proteins by an anti-Larp7 antibody directed against the N-terminal half of Larp7. **d** Co-immunoprecipitation of proteins by anti-Smn from NSC-34 cells. Lack of Gapdh co-precipitation serves as specificity control. **e** Sucrose gradient fractionation of NSC-34 cell lysate. Individual fractions were probed by Western blotting for proteins indicated on the right. Lower panel: Detection of 7SK in each fraction by qPCR. a.u., arbitrary units. **f** Co-immunoprecipitation of proteins by anti-SmB/B′ from NSC-34 cells. **g** qPCR analysis of 7SK co-precipitated by anti-SmB/B′ from NSC-34 cells. Data are mean ± standard deviation (s.d.); ***$P \leq$ 0.001; unpaired two-tailed *t*-test (*n* = 3). **h** Co-immunoprecipitation of proteins by anti-SmD1 from NSC-34 cells. Source data are provided as a Source Data file.

7SK pulldown data (Supplementary Fig. 1d). These RNA-binding proteins were previously identified as components of 7SK/hnRNP particles[24]. Gene ontology (GO) term enrichment analysis revealed that components of the Smn complex were enriched in the Biotin-pd7SK co-precipitate (Supplementary Fig. 1e and

Supplementary Data 3). Among these proteins were Smn, Gemins, and Sm proteins (Fig. 1a). This finding suggests that 7SK/hnRNP particles interact with the Smn complex (Supplementary Fig. 1f).

To further investigate this interaction, we immunoprecipitated Larp7, an essential component of 7SK particles[13], from NSC-34

cell lysate and assessed co-purified proteins by immunoblot analysis. Immunoprecipitation with antibodies against endogenous Larp7 co-purified Smn, hnRNP R, and hnRNP A1 (Fig. 1b, c). 7SK was highly enriched in the Larp7 co-precipitate indicating specificity of the procedure (Supplementary Fig. 2a). In the human SMN complex, SMN interacts closely with GEMIN2[29]. We found that Gemin2 was co-purified with Larp7 from NSC-34 lysate (Fig. 1c) indicating that both Smn and Gemin2 interact with 7SK particles. Smn and Gemin2 were also co-purified using an antibody against Mepce, another component of the 7SK core RNP (Supplementary Fig. 2b). In addition to NSC-34 cells, we observed an interaction of SMN and GEMIN2 with LARP7 and MePCE also in HEK293TN and HeLa cells (Supplementary Fig. 2c–f). Beyond Smn and Gemin2, Gemin3-5 were significantly enriched by 7SK pulldown (Fig. 1a). In agreement, we observed co-precipitation of GEMIN3 and GEMIN4 with anti-LARP7 and anti-MePCE (Supplementary Fig. 2c, d, g–l). Given that SMN is stably associated with GEMIN2, and GEMIN3 with GEMIN4 and 5[29], this suggests that 7SK particles are associated with SMN/GEMIN2 and GEMIN3/4/5 subcomplexes.

We further analyzed the association between 7SK and Smn complexes by immunoprecipitation with an antibody against Smn. This co-purified Larp7 (Fig. 1d) and 7SK RNA (Supplementary Fig. 2m). Importantly, hnRNP R and hnRNP A1, but not Hexim1 were co-purified with anti-Smn (Fig. 1d). This indicates that Smn interacts with 7SK/hnRNP but not 7SK/Hexim1/P-TEFb complexes. In agreement with this notion, immunoprecipitation of Cyclin T1 co-precipitated Hexim1 alongside Larp7 and 7SK, but not hnRNP R, hnRNP A1 or Smn (Supplementary Fig. 2n, o). In addition to hnRNP R and hnRNP A1, other RNA-binding proteins have previously been identified as 7SK interactors including hnRNP Q and RHA[24,25]. In line with these studies, we observed co-purification of hnRNP Q and RHA with anti-LARP7 and anti-MePCE (Supplementary Fig. 2g–l). SMN has previously been shown to interact with RHA[30]. In agreement, we found that RHA co-immunoprecipitated with SMN in NSC-34, HeLa, and HEK293TN cells (Supplementary Fig. 2p–r). Taken together, our results indicate that Smn associates with 7SK complexes containing hnRNP A1 and R, and that hnRNP Q and RHA are also part of complexes involving 7SK and Smn.

We then probed whether the Smn interactions with 7SK components are dependent on RNA. For this purpose we carried out immunoprecipitations on lysates pretreated with RNase. We found that co-immunoprecipitation of hnRNP R and hnRNP A1 with Smn was abolished by RNase treatment, whereas Larp7 and Mepce were still co-purified with Smn in the absence of RNA (Supplementary Fig. 3a, b). Similarly, immunoprecipitation of Larp7 or Mepce co-purified Smn and Gemin2 independent of RNA, while co-purification of hnRNP R, hnRNP A1 and, to a lesser extent, Cyclin T1 and Cdk9 was perturbed by RNase treatment (Supplementary Fig. 3c–e). Thus, the interactions of the Smn complex with hnRNP proteins are mediated through RNA whereas its association with Larp7 and Mepce involves protein–protein contacts.

Next, we investigated the association of 7SK and Smn complexes by sucrose density gradient ultracentrifugation of NSC-34 cell lysate (Fig. 1e). Smn and Gemin2 co-sedimented across most of the gradient as has been shown previously[29], with higher levels being present in the bottom fractions. Interestingly, hnRNP R sedimentation was similar to Smn and Gemin2 whereas hnRNP A1 was mostly present in lighter fractions indicating that hnRNP R is more closely associated with the Smn complex than hnRNP A1. Even though Larp7 and 7SK were distributed at the top of the gradient in fractions 3–9, their sedimentation profiles partially overlapped with that of Smn, Gemin2, hnRNP R, and hnRNP A1. On the other hand, Hexim1 and Cdk9 co-sedimented very similar to Larp7

indicating their tight association with the 7SK complex. Together with our co-immunoprecipitation results, this finding suggests that under steady-state conditions a fraction of 7SK is associated with Smn complexes and hnRNP R.

We also analyzed the sedimentation profile of SmB/B', a substrate of the Smn complex for the assembly of snRNPs. SmB/B' had a distribution very similar to Larp7 and was detectable in fractions 2–9 (Fig. 1e). Immunoprecipitation of SmB/B' from NSC-34 lysate using an anti-SmB/B' antibody[31] co-purified Larp7 in addition to Smn and Gemin2 (Fig. 1f). Vice versa, SmB/B' was co-purified with antibodies against Larp7 and Mepce (Fig. 1c and Supplementary Fig. 2b–f). We also observed an association of 7SK with SmB/B' by RNA immunoprecipitation (Fig. 1g). The anti-SmB/B' antibody efficiently co-purified U1 snRNA showing its specificity (Supplementary Fig. 3f). These data suggest that SmB/B' associates with 7SK particles. Similar to Smn (Supplementary Fig. 3a, b), SmB/B' interacted with Larp7 and Mepce in an RNA-independent manner whereas its binding to hnRNP R and hnRNP A1 was abolished by RNase treatment (Supplementary Fig. 3g, h). Likewise, immunoprecipitation with anti-Larp7 or anti-Mepce co-purified SmB/B' independent of RNA (Supplementary Fig. 3c, d). Beyond SmB/B', we also observed an association of Larp7 with SmD1 (Fig. 1h). Together, our data suggest that the Smn complex and Sm proteins interact with 7SK particles.

**Transcription regulates the association of Smn with 7SK complexes.** The composition of 7SK subcomplexes has been shown to depend on the transcriptional activity of a cell[24,25]. In order to investigate whether the interaction of Smn with 7SK complexes depends on transcription, we treated NSC-34 cells with the transcriptional inhibitor Actinomycin D (ActD) for 1 h and performed anti-Larp7 immunoprecipitations. Following transcriptional inhibition, we observed enhanced association of Smn, Gemin2, and SmB/B' with Larp7 (Fig. 2a). In contrast, Hexim1 and Cdk9 binding to Larp7 was reduced by ActD treatment (Fig. 2a). Enhanced association of Larp7 with Smn upon ActD exposure was also detectable by anti-Smn immunoprecipitation (Fig. 2b). We further confirmed these findings by immunoprecipitating Mepce, which co-purified increased levels of Smn, SmB/B' and hnRNP R and reduced levels of Hexim1 and Cdk9 following ActD treatment (Fig. 2c, d). Also, the association of the Smn interactor Gemin2 with Mepce was enhanced upon ActD exposure (Fig. 2d). Next, we investigated whether the reorganization of 7SK complexes also occurs in cultured embryonic mouse spinal motoneurons. Similar to NSC-34 cells, the association of Larp7 with Smn was enhanced in motoneurons by ActD treatment (Fig. 2e). Likewise, increased binding of MePCE and LARP7 to SMN was detectable in HEK293TN cells treated with ActD for 1 or 6 h (Fig. 2f). In contrast, GEMIN2 binding to SMN was unperturbed by ActD exposure. Taken together, these data indicate that the association of the Smn complex with 7SK/hnRNP particles is enhanced when transcription is reduced.

**Reduced snRNP assembly upon transcriptional inhibition.** Given that the Smn complex has an essential function in snRNP biogenesis, and that transcriptional inhibition enhances the association between the Smn complex and 7SK particles, we next asked whether snRNP production was modified by ActD treatment. For this purpose we first investigated the effect of ActD exposure time on the interaction between Smn and 7SK complexes. We focused on Larp7 and Mepce, which we found to interact with Smn in an RNA-independent manner. ActD treatment of NSC-34 cells for 0, 1, 6, and 12 h reduced total levels of Larp7 protein relative to DMSO control treatment while Smn

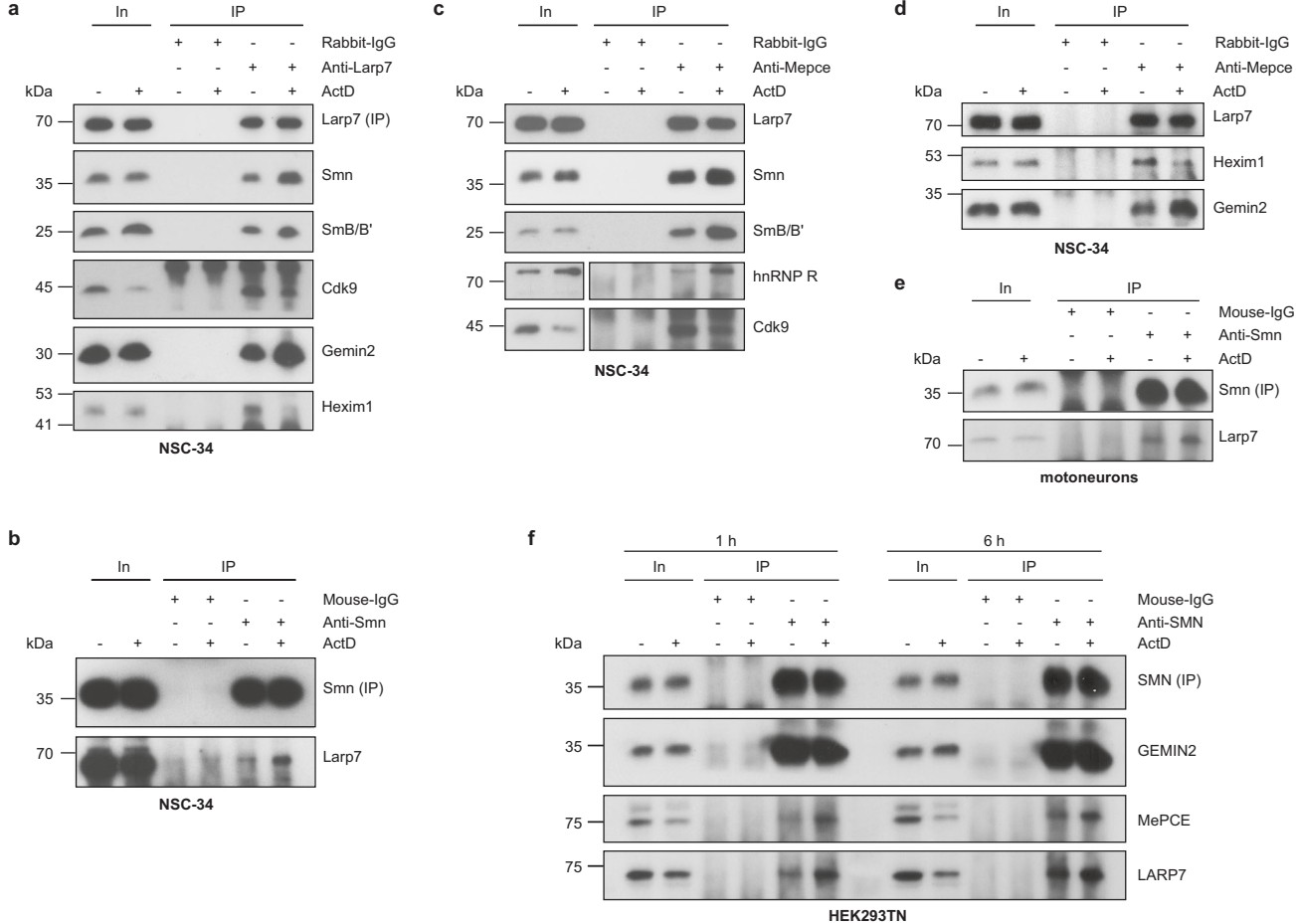

**Fig. 2 Transcription-dependent regulation of 7SK/hnRNP/Smn complexes. a** Western blot analysis of proteins co-immunoprecipitated by anti-Larp7 from NSC-34 cells treated with DMSO or actinomycin D (ActD) for 1 h. **b** Co-immunoprecipitation of Larp7 by anti-Smn from NSC-34 cells treated with DMSO or ActD for 1 h. **c, d** Co-immunoprecipitation of proteins by anti-Mepce from NSC-34 cells treated with DMSO or ActD for 1 h. For Western blots of hnRNP R and Cdk9, a shorter exposure of the inputs was chosen. **e** Co-immunoprecipitation of Larp7 by anti-Smn from cultured mouse primary motoneurons treated with DMSO or ActD for 6 h. **f** Co-immunoprecipitation of proteins by anti-SMN from HEK293TN cells treated with DMSO or ActD for 1 or 6 h. Source data are provided as a Source Data file.

protein levels remained unchanged (Fig. 3a, b). At the same time, increasing amounts of Larp7 co-purified with Smn with increasing duration of ActD treatment, reaching a peak at 6 h (Fig. 3c, d). At this time point, the association of Mepce with Smn (Fig. 3e) and with SmB/B' (Fig. 3f) was also enhanced by ActD relative to DMSO treatment.

Next, we analyzed total levels of Sm-class snRNAs by qPCR following ActD exposure (Fig. 3g). In line with the activity of ActD as a transcriptional inhibitor, we observed reduced snRNA levels upon ActD treatment relative to DMSO control. Nevertheless, we found that the amounts of individual snRNAs were differentially affected by transcriptional inhibition. U4atac and U11 showed the strongest decrease in levels after 6 and 12 h of ActD treatment relative to DMSO control. The levels of U1, U2, and U5 were also reduced at these time points, albeit to a lesser extent. In contrast, the amounts of U4 and U12 were unaffected or even increased over a period of 12 h of ActD treatment. In order to quantify snRNP levels we performed RNA immunoprecipitation with anti-SmB/B' followed by qPCR detection of snRNAs in the co-precipitate. We found that the association of U2 and U12 with SmB/B' was significantly reduced after 6 and 12 h of ActD treatment (Fig. 3h). Thus, transcriptional inhibition not only leads to reduced snRNA levels but also to reduced amounts of selected snRNPs. We then assessed snRNP assembly

using an in vitro snRNP assembly assay[32,33]. Biotinylated U2 snRNA (Fig. 3i) was incubated with cytosolic extracts from DMSO-treated or ActD-treated NSC-34 cells in the presence or absence of ATP. Smn-mediated snRNP assembly was monitored by pulldown with streptavidin beads and immunoblot analysis of SmB/B' binding (Fig. 3j). We found that addition of ATP stimulated snRNP assembly as demonstrated previously[32]. Compared to control cells treated with DMSO, snRNP assembly was significantly reduced in cells exposed to ActD (Fig. 3j, k). Thus, snRNP production is downregulated upon transcriptional inhibition.

**Smn interacts with 7SK/hnRNP complexes in the nucleus and cytosol.** Smn exerts its cellular function in snRNP biogenesis mostly in the cytosol[4]. In order to identify the subcellular compartment in which Smn interacts with 7SK particles, we used digitonin to separate cytosolic proteins (Cyt fraction) from organelles and nuclear soluble proteins not associated with chromatin (Nuc fraction), and also from chromatin-associated proteins (Chr fraction)[34]. To investigate the specificity of the fractionation procedure we first measured the distribution of marker proteins by immunoblotting. We observed that Gapdh as a cytosolic marker was present almost exclusively in the Cyt fraction,

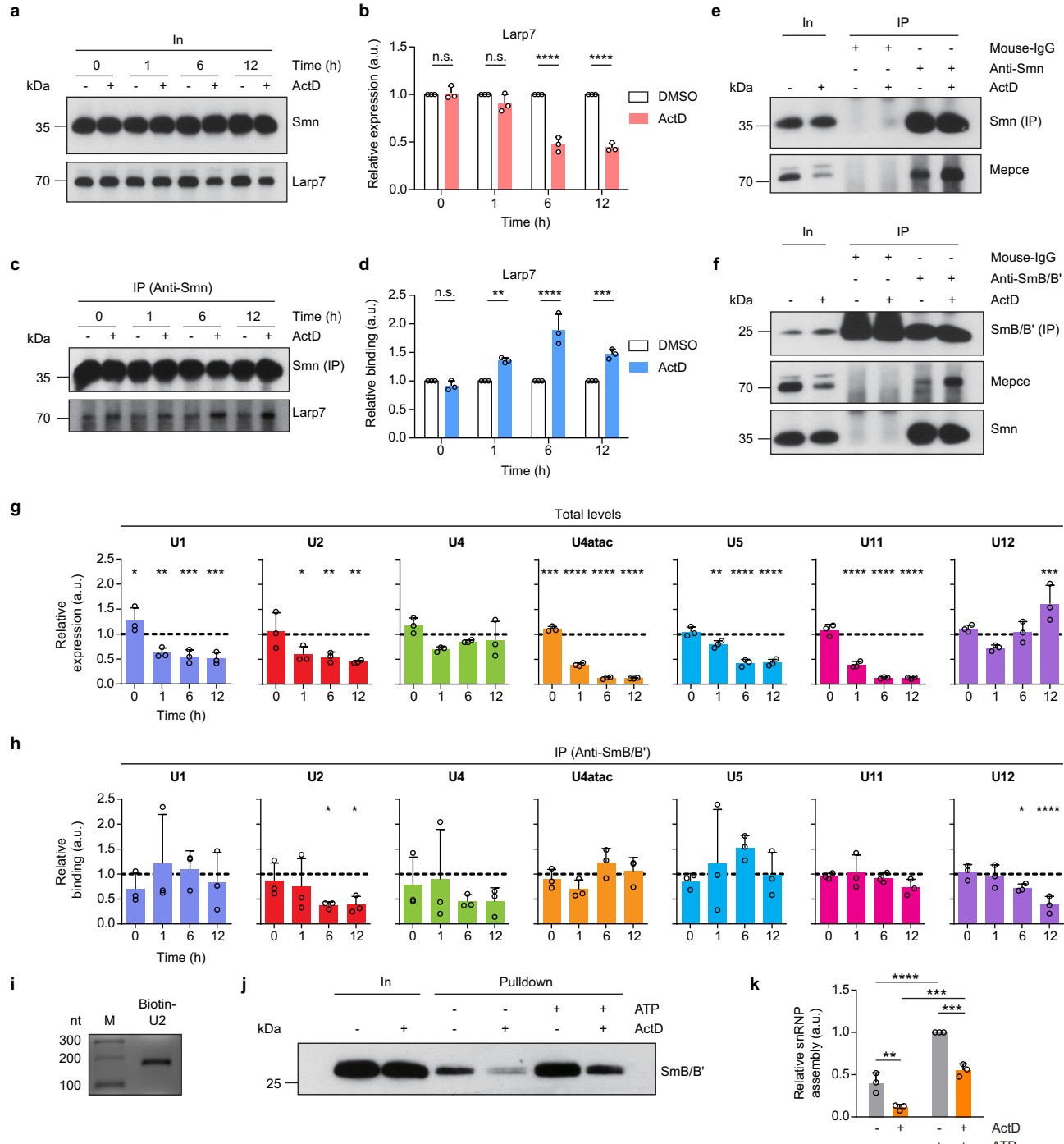

Calnexin as organellar marker in the Nuc fraction and Histone H3 as marker for chromatin in the Chr fraction (Fig. 4a, b).

Next we investigated the distribution of 7SK complex components (Fig. 4a, b). Larp7 was present mostly in the Nuc and Chr fractions reflecting its abundance in the nucleus, but also at relatively high levels in the Cyt fraction. The distribution of Larp7 was closely mirrored by 7SK (Fig. 4c). hnRNPs were abundant in the nuclear fractions, especially the Chr fraction containing chromatin. This distribution reflects their association with nascent RNAs and roles in pre-mRNA processing. Nevertheless, hnRNPs were also detectable in the Cyt fraction. In agreement with immunostaining data in primary motoneurons[35], Smn was detectable mostly in the cytosolic Cyt fraction. To substantiate these findings we investigated the distribution of 7SK

interactors in subcellular fractions of primary mouse motoneurons (Supplementary Fig. 4a). Similar to NSC-34 cells, hnRNP R was mostly present in the nuclear fractions but also detectable in the Cyt fraction while Smn was abundant in the cytosol. The 7SK core component Larp7 was abundant in the Nuc fraction but also localized to the cytosolic fraction in agreement with our previous data detecting 7SK in the cytosol of motoneurons by in situ hybridization[26]. To confirm this finding we investigated the distribution of Larp7 in motoneurons by immunostaining (Supplementary Fig. 4b). Larp7 immunoreactivity was highest in the nucleus but also detectable in the cytoplasm. The Larp7 immunosignal was strongly reduced when Larp7 was knocked down via lentiviral delivery of a short hairpin RNA (shRNA) targeting the *Larp7* transcript, thereby showing specificity of the

**Fig. 3 Reduced snRNP production upon transcriptional inhibition. a** Western blot analysis of Smn and Larp7 protein levels in NSC-34 cells treated with DMSO or ActD for the indicated durations. **b** Quantification of Larp7 protein levels in **a**. Data are mean ± s.d.; ****$P \leq 0.0001$; n.s., not significant; two-way ANOVA with Sidak's multiple-comparisons test ($n = 3$). **c** Co-immunoprecipitation of Larp7 by anti-Smn from NSC-34 cells treated with DMSO or ActD for the indicated durations. **d** Quantification of Larp7 co-purification by anti-Smn in **c**. Data are mean ± s.d.; **$P \leq 0.01$; ***$P \leq 0.001$; ****$P \leq 0.0001$; n.s., not significant; two-way ANOVA with Sidak's multiple-comparisons test ($n = 3$). **e** Co-immunoprecipitation of Mepce by anti-Smn from NSC-34 cells treated with DMSO or ActD for 6 h. **f** Co-immunoprecipitation of Mepce and Smn by anti-SmB/B' from NSC-34 cells treated with DMSO or ActD for 6 h. Note that the immunosignal at 25 kDa for mouse-IgG immunoprecipitation is non-specific from the antibody light chain. **g** qPCR analysis of total snRNA levels in NSC-34 cells treated with DMSO or ActD for the indicated durations. Data are mean ± s.d.; *$P \leq 0.05$; **$P \leq 0.01$; ***$P \leq 0.001$; ****$P \leq 0.0001$; two-way ANOVA with Sidak's multiple-comparisons test ($n = 3$). **h** qPCR analysis of snRNAs co-precipitated by anti-SmB/B' from NSC-34 cells treated with DMSO or ActD for the indicated durations. Data are mean ± s.d.; *$P \leq 0.05$; ****$P \leq 0.0001$; two-way ANOVA with Sidak's multiple-comparisons test ($n = 3$). **i** Agarose gel electrophoresis of biotinylated U2 snRNA. **j** In vitro snRNP assembly assay. Biotinylated U2 snRNA was incubated with cytosolic extracts from NSC-34 cells exposed to DMSO or ActD for 6 h. Reactions were assembled with or without exogenous ATP. Biotinylated U2 snRNPs were pulled down with streptavidin beads and SmB/B' was analyzed by Western blot. **k** Quantification of in vitro U2 snRNP assembly assay in **j**. Data are mean ± s.d.; **$P \leq 0.01$; ***$P \leq 0.001$; ****$P \leq 0.0001$; two-way ANOVA with Tukey's multiple-comparisons test ($n = 3$). Source data are provided as a Source Data file.

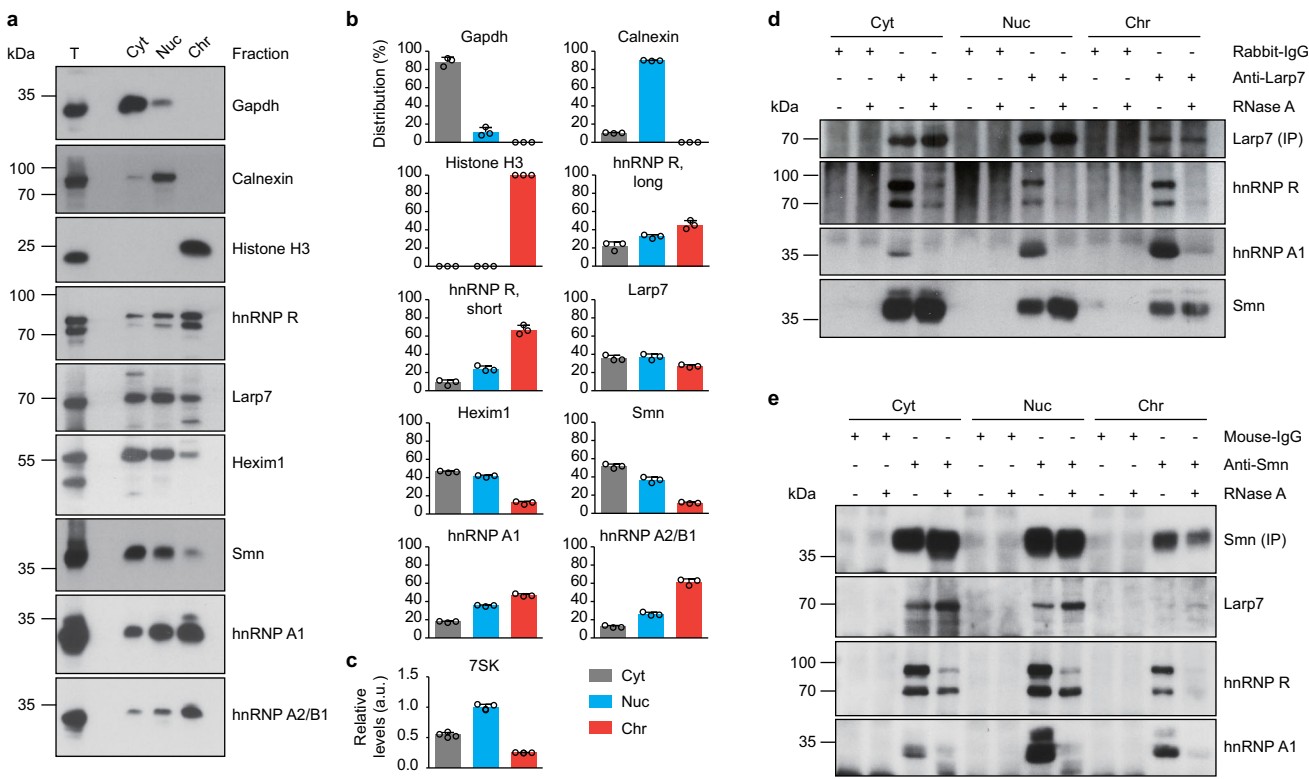

**Fig. 4 Smn interacts with 7SK/hnRNP complexes in the nucleus and cytosol. a** Western blot analysis of NSC-34 subcellular fractions. Cyt, cytosol; Nuc, nuclear soluble proteins and organelles; Chr, chromatin-associated proteins; T, total lysate. **b** Quantification of Western blot signals in **a**. Data are mean ± s.d. ($n = 3$). **c** Quantification of 7SK RNA levels by qPCR. Data are mean ± s.d. ($n = 4$). **d** Western blot analysis of proteins co-immunoprecipitated by anti-Larp7 from subcellular fractions of NSC-34 cells. Fractions were pretreated with RNase as indicated. **e** Same as in **d** but with anti-Smn. Source data are provided as a Source Data file.

immunostaining procedure. Thus, components of the 7SK complex are not only present in the nucleus but also localize to the cytosol.

To investigate whether Smn interacts with 7SK complexes in the nucleus or cytosol we performed immunoprecipitations from each NSC-34 subcellular fraction. Immunoprecipitation with anti-Larp7 showed that the association of Larp7 with hnRNP R and Smn was most prominent in the Cyt fraction, while its interaction with hnRNP A1 was more detectable in the nuclear fractions (Fig. 4d). To assess RNA dependence of these interactions we performed immunoprecipitations on fractions pretreated with RNase. Similar to our immunoprecipitation experiments with whole cell lysates, we observed that co-purification of hnRNP R and hnRNP A1 with Larp7 was

abolished by RNase treatment, while the association of Smn with Larp7 was resistant to RNase. We obtained an analogous result using an antibody against Smn for immunoprecipitation (Fig. 4e). Taken together, our results show that 7SK complexes containing hnRNP R associate with Smn in the cytosol.

**Smn interacts with hnRNP R via 7SK.** Next, we analyzed the interaction between Smn and 7SK/hnRNP R complexes. To do so, we knocked down 7SK or Larp7 using shRNAs delivered into NSC-34 cells via lentiviral transduction (Supplementary Fig. 5a) and performed anti-Smn immunoprecipitations. Under control conditions, Larp7, hnRNP R, and hnRNP A1, but not Hexim1, co-purified with Smn (Supplementary Fig. 5b). Upon loss of 7SK, co-purification of hnRNP R but not hnRNP A1 was reduced

(Supplementary Fig. 5b). In contrast, knockdown of Larp7 did not affect co-immunoprecipitation of hnRNP R with Smn (Supplementary Fig. 5b). Likewise, co-purification of hnRNP R with SmB/B' was reduced in 7SK knockdown but not in Larp7 knockdown cells (Supplementary Fig. 5c). We did not observe an association of hnRNP A1 with SmB/B' under the control and knockdown conditions suggesting that the interaction of SmB/B' with hnRNP A1 is less stable compared to that with hnRNP R. While both hnRNP R and hnRNP A1 are associated with 7SK, it is possible that they form individual 7SK/hnRNP A1 and 7SK/hnRNP R subcomplexes that differ with respect to their affinity for SmB/B'.

Our finding that the association of hnRNP R with Smn and SmB/B' was reduced by 7SK knockdown, but not by Larp7 knockdown is surprising given that knockdown of Larp7 also depleted 7SK in agreement with its function in stabilizing 7SK (Supplementary Fig. 5a). Therefore, we tested whether the association between Smn and hnRNP R remained RNase-sensitive following loss of Larp7. Indeed, co-purification of hnRNP R with Smn from Larp7 knockdown cells was abolished upon RNase treatment (Supplementary Fig. 5d). This indicates that other RNAs beyond 7SK might be associated with Larp7 that can contribute to the interaction between Smn and hnRNP R. One such candidate RNA would be β-actin mRNA (Actb), which associates with hnRNP R in a Smn-dependent manner[36]. RNA immunoprecipitation with anti-Larp7 co-purified high amounts of β-actin mRNA alongside 7SK (Supplementary Fig. 5e). To determine the specificity of the association of Larp7 and β-actin mRNA, we analyzed the presence of snRNAs in the Larp7 co-precipitate. Among these, U6 snRNA has previously been shown to interact with Larp7, albeit with lower efficiency compared to 7SK[13,15]. In agreement with these studies, we observed an association of Larp7 with U6 snRNA, which was much weaker than Larp7 binding to 7SK and β-actin mRNA (Supplementary Fig. 5e). We also observed co-purification of β-actin mRNA when we used an antibody against Mepce for immunoprecipitation (Supplementary Fig. 5f). This indicates that β-actin mRNA associates with 7SK particles containing Mepce and Larp7, and thus might contribute to the interaction of Smn with 7SK/hnRNP R complexes.

We also investigated which RNA regions of 7SK are important for its interaction with Smn and hnRNP R. To do so, we performed RNA pulldown experiments using 7SK RNA prepared by in vitro transcription. We used wildtype (WT) 7SK and its antisense (AS) sequence as control. Additionally, we generated 7SK RNAs with deletions of SL1, 2, 3, or 1–3 (Supplementary Fig. 6a). SL4, which is known to bind Larp7, and a region of SL1 bound by Mepce[15] were preserved in all constructs. We used biotinylated oligonucleotides to couple the 7SK RNAs to Streptavidin-coated beads (Supplementary Fig. 6b). Following incubation with NSC-34 cell lysate, we observed that Larp7, hnRNP R, Smn, and hnRNP A1 bound to 7SK WT but not to 7SK AS control (Supplementary Fig. 6c). In agreement with our previous study[26], deletion of SL3 reduced binding of hnRNP R to 7SK. In contrast, the interaction of Smn with 7SK was unaffected by deletion of SL3. Instead, Smn binding to 7SK/Larp7 was abolished in the ΔSL123 mutant lacking SL1-3. This finding is somewhat surprising given that Smn interacts with Larp7 through protein contacts (Supplementary Fig. 3a), and that Larp7 binding to 7SK is unaffected by the ΔSL123 deletion (Supplementary Fig. 6c). One possibility is that deletion of SL1-3 might place Larp7 in close proximity to Mepce bound at the 5′ end of ΔSL123, thereby preventing Smn binding due to steric hindrance. To investigate this possibility we generated a chimeric 7SK RNA in which we replaced SL1–3 with an unrelated sequence from GFP (ΔSL123-GFP), and used it as bait for pulldown (Supplementary

Fig. 6a, d). We found that Smn binding to ΔSL123-GFP was similar to its binding to 7SK WT (Supplementary Fig. 6e). These results are in agreement with a model according to which Smn binding to Larp7 and Mepce occurs independent of RNA while hnRNP R binds to 7SK through SL3.

**Components of the Smn complex regulate its association with 7SK.** Our data indicate that transcriptional inhibition enhances the association of the Smn complex with 7SK RNPs and down-regulates snRNP production (Figs. 2 and 3). Based on this finding we next investigated whether inhibition of snRNP assembly itself has an impact on the interaction of the Smn complex with 7SK particles. For this purpose we knocked down SmB/B', one of the substrate proteins for the production of snRNPs, in NSC-34 cells and performed co-immunoprecipitation experiments with an antibody against Larp7. Compared to control cells, SmB/B' depletion increased the co-purification of Smn and Gemin2 with Larp7 (Fig. 5a, b). In contrast, the association of Hexim1 with Larp7 was decreased in SmB/B' knockdown cells (Fig. 5a, b). To corroborate these findings, we knocked down Gemin2, which stably interacts with SMN and is required for Sm protein binding to the SMN complex for snRNP production[29,37]. Similar to SmB/B' depletion, we observed increased association of Smn with Larp7 upon knockdown of Gemin2 (Fig. 5c). Thus, blocking of snRNP production causes dissociation of 7SK/Hexim1/P-TEFb complexes and increased association of Smn and other components of the Smn complex with 7SK particles.

We also assessed whether reduction of Smn itself perturbs the association of other components of the Smn complex with 7SK particles. When we knocked down Smn in NSC-34 cells, we observed reduced association of Gemin2 and SmB/B' with Larp7 (Fig. 5d). However, co-purification of Hexim1 with Larp7 was unperturbed by Smn depletion (Fig. 5e). This suggests that the abundance of 7SK/Hexim1/P-TEFb complexes is not altered by loss of Smn. Co-purification of Larp7 with SmB/B' was also reduced in brains from Smn−/−;SMN2 mice, an established mouse model for SMA type 1[38], compared to Smn+/+;SMN2 controls (Fig. 5f). Of note, while we detected co-precipitation of hnRNP R with SmB/B' we did not observe an association of hnRNP A1 with SmB/B' in mouse brain. This agrees with our previous result in NSC-34 cells (Supplementary Fig. 5c) and further indicates that hnRNP A1 is less stably associated with SmB/B'. Taken together, our results suggest that the interaction of the Smn complex with 7SK particles involves the Smn protein directly (Fig. 5g).

**7SK particles regulate Smn-mediated snRNP biogenesis.** We then assessed how 7SK particles regulate Smn functions in snRNP assembly. We first tested the possibility that transcriptional inhibition disrupts the composition of Smn complexes. For this purpose, we immunoprecipitated Smn from NSC-34 cells treated with ActD or DMSO control for 6 h and assessed Gemin2 binding. While the association of Larp7 with Smn was enhanced upon ActD exposure, co-purification of Gemin2 with Smn was unchanged by transcriptional inhibition (Fig. 6a). Thus, interaction with 7SK particles does not disrupt binding of Smn to its key partner, Gemin2. Next, we investigated the subcellular site where 7SK regulates the activity of the Smn complex. We immunoprecipitated Smn from subcellular fractions of NSC-34 cells exposed to ActD or DMSO and assessed co-purification of Larp7 (Fig. 6b). Following ActD treatment, Smn interaction with Larp7 was increased in the Cyt fraction and, to a lesser extent, in the Chr fraction while it was decreased in the Nuc fraction. We also used RNA immunoprecipitation to investigate whether the binding of 7SK RNA to the Smn complex is altered by ActD

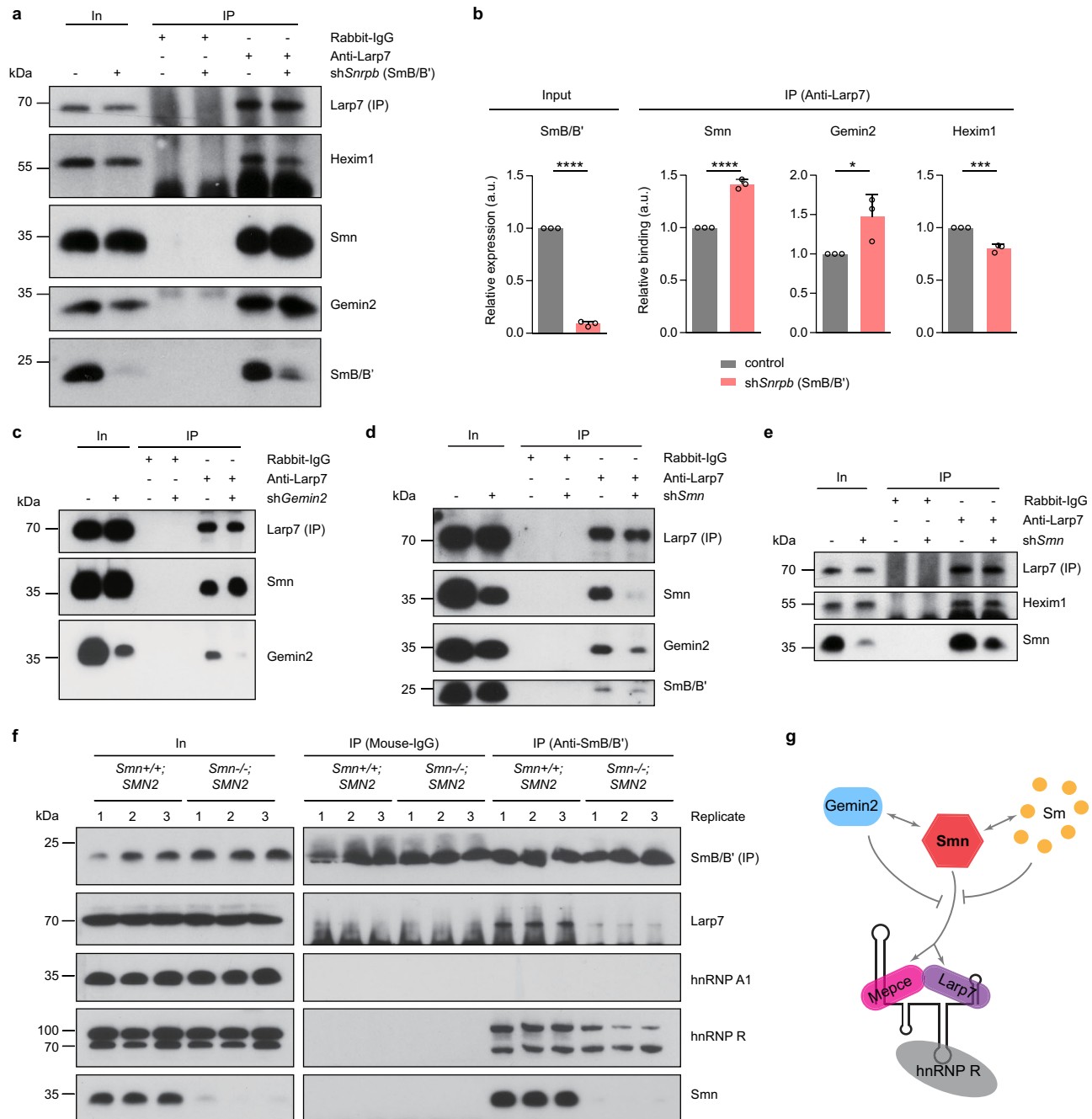

**Fig. 5 Depletion of individual components of the Smn complex alters its binding to Larp7. a** Western blot analysis of proteins co-immunoprecipitated by anti-Larp7 from control and SmB/B′ knockdown (sh*Snrpb*) NSC-34 cells. **b** Quantification of Western blot signals in **a**. Data are mean ± s.d.; *$P \le 0.05$; ***$P \le 0.001$; ****$P \le 0.0001$; unpaired two-tailed t-test (n = 3). **c** Co-immunoprecipitation of proteins by anti-Larp7 from control or Gemin2 knockdown (sh*Gemin2*) NSC-34 cells. **d,e** Co-immunoprecipitation of proteins by anti-Larp7 from control or Smn knockdown (sh*Smn*) NSC-34 cells. **f** Co-immunoprecipitation of proteins by anti-SmB/B′ from brains of $Smn^{+/+};SMN2$ or $Smn^{-/-};SMN2$ mice. Note that the immunosignal at 25 kDa for mouse-IgG immunoprecipitation is non-specific from the antibody light chain. **g** Model of Smn interactions with 7SK/hnRNP R. Source data are provided as a Source Data file.

exposure. We found that ActD treatment increased the amounts of 7SK co-precipitating with SmB/B′ (Fig. 6c) or Smn (Fig. 6d). In contrast, the association of U1 or U2 snRNA with Smn was unchanged by transcriptional inhibition (Fig. 6d). These results indicate that 7SK particles regulate the Smn complex in the cytosol without disrupting its composition.

To further investigate the functional significance of the interaction of Larp7 and Mepce with Smn for regulation of snRNP production, we knocked down Larp7 and Mepce

simultaneously in NSC-34 cells and investigated snRNP levels following ActD exposure. Lentiviral transduction with shRNAs reduced Larp7 protein levels by >75% and Mepce protein levels by >85% (Fig. 6e, f). In order to detect snRNPs, we immunoprecipitated SmB/B′ as before and measured snRNAs in the co-precipitate by qPCR. For this analysis we chose U1 snRNA, whose association with SmB/B′ was not altered by ActD exposure, and U2 snRNA, which bound less to SmB/B′ following ActD treatment (Fig. 3h). In control cells, we detected

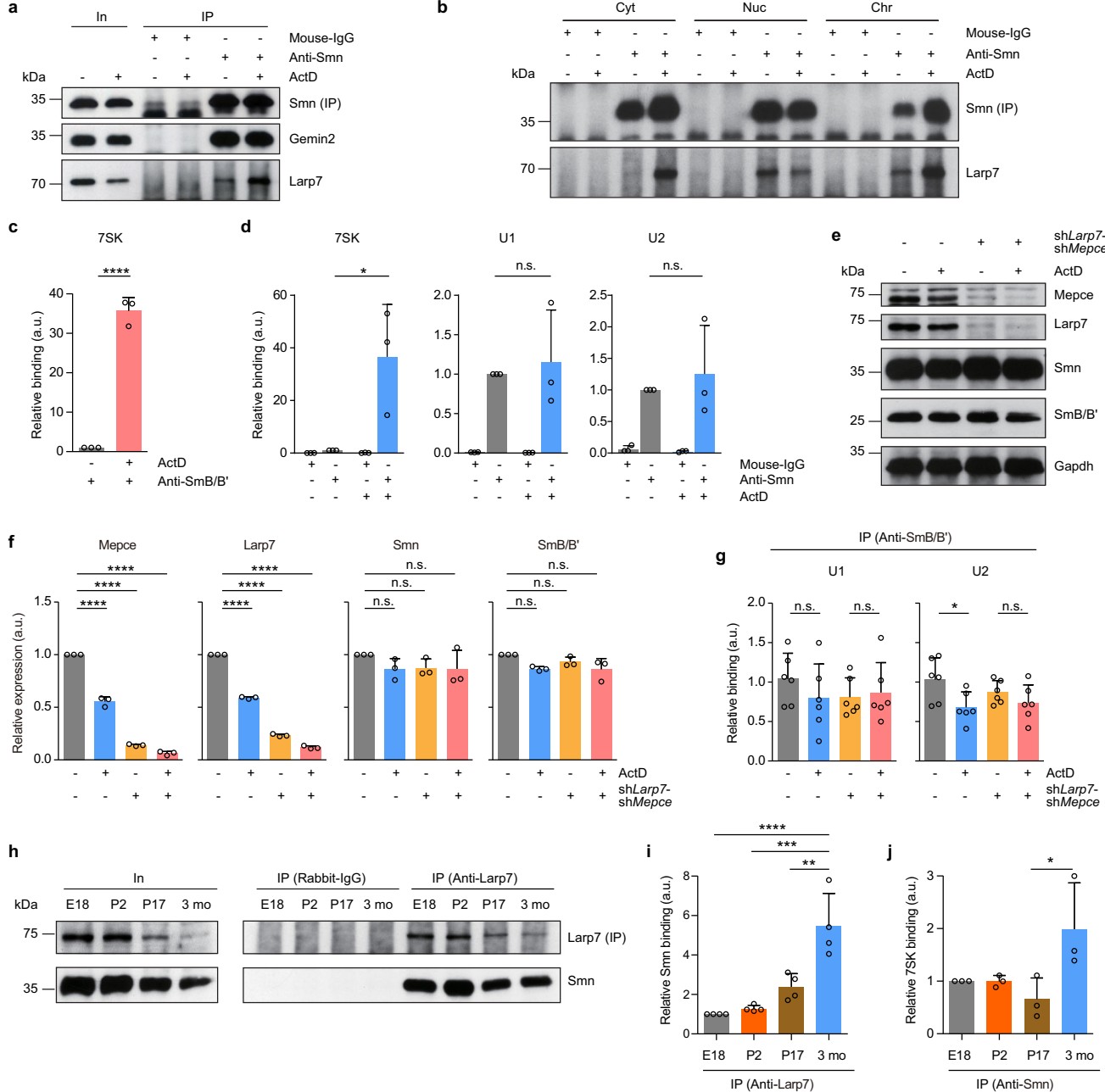

**Fig. 6 The 7SK complex regulates Smn complex activity. a** Western blot analysis of proteins co-immunoprecipitated by anti-Smn from NSC-34 cells treated with DMSO or ActD for 6 h. **b** Co-immunoprecipitation of Larp7 by anti-Smn from subcellular fractions of NSC-34 cells treated with DMSO or ActD for 1 h. Cyt, cytosol; Nuc, nuclear soluble proteins and organelles; Chr, chromatin-associated proteins. **c** qPCR analysis of 7SK co-precipitated by anti-SmB/B' from NSC-34 cells treated with DMSO or ActD for 6 h. Data are mean ± s.d.; ****$P \leq 0.0001$; unpaired two-tailed $t$-test ($n = 3$). **d** qPCR analysis of 7SK and snRNAs co-precipitated by anti-Smn from NSC-34 cells treated with DMSO or ActD for 6 h. Data are mean ± s.d.; *$P \leq 0.05$; n.s., not significant; two-way ANOVA with Tukey's multiple-comparisons test ($n = 3$). **e** Western blot analysis of Larp7/Mepce double-knockdown (sh*Larp7*-sh*Mepce*) or control NSC-34 cells treated with DMSO or ActD for 6 h. **f** Quantification of Western blot signals in **e**. Data are mean ± s.d.; ****$P \leq 0.0001$; n.s., not significant; two-way ANOVA with Tukey's multiple-comparisons test ($n = 3$). **g** qPCR analysis of U1 and U2 snRNA co-precipitated by anti-SmB/B' from Larp7/Mepce double-knockdown or control NSC-34 cells treated with DMSO or ActD for 6 h. Data are mean ± s.d.; *$P \leq 0.05$; n.s., not significant; two-way ANOVA with Tukey's multiple-comparisons test ($n = 6$). **h** Co-immunoprecipitation of Smn by anti-Larp7 from brains of mice aged E18, P2, P17, and 3 months. **i** Quantification of Western blot signals in **h**. Data are mean ± s.d.; **$P \leq 0.01$; ***$P \leq 0.001$; ****$P \leq 0.0001$; one-way ANOVA with Tukey's multiple-comparisons test ($n = 4$). **j** qPCR analysis of 7SK RNA co-precipitated by anti-Smn from mouse brains. Data are mean ± s.d.; *$P \leq 0.05$; one-way ANOVA with Tukey's multiple-comparisons test ($n = 3$). Source data are provided as a Source Data file.

reduced U2 snRNP levels upon ActD exposure whereas U1 snRNP levels were unchanged (Fig. 6g). This reduction in U2 snRNP levels by ActD treatment was abolished in cells depleted of Larp7 and Mepce (Fig. 6g). Thus, Larp7 and Mepce,

whose interaction with Smn is enhanced by transcriptional inhibition, regulate the cellular capacity for snRNP production.

The snRNP assembly activity of the Smn complex in the nervous system is high at embryonic stages and then reduced

postnatally[11]. In mouse brain, this decrease in snRNP assembly occurs at 2–3 weeks of age. In this context, we investigated whether the association between the 7SK component Larp7 and Smn is developmentally regulated in mouse brain. We prepared brain lysates of mice aged E18, P2, P17, and 3 months and immunopurified Larp7. At all ages, Smn co-purified with Larp7 (Fig. 6h). Quantification of Smn co-purification with Larp7- showed a significant increase in binding between P17 and 3 months of age (Fig. 6i). Between these time points we also observed increased association of 7SK with Smn (Fig. 6j). Thus, the decrease in snRNP assembly during brain development is accompanied by increased association between Smn and 7SK complexes in agreement with a function of the 7SK complex in regulating Smn activity.

## Discussion

Here, we provide evidence that the Smn complex, which mediates spliceosomal snRNP production, associates with and is regulated by the 7SK complex. We report that Smn interacts with the 7SK core components Larp7 and Mepce. This interaction is enhanced upon transcriptional inhibition, leading to reduced snRNP levels. Thus, our data extend the current model of 7SK function and suggest that the transcriptional activity of a cell not only determines the balance between 7SK/Hexim1/P-TEFb and 7SK/hnRNP complexes in the nucleus, but also controls snRNP production by regulating the association of the Smn complex with 7SK particles in the cytosol.

We found that 7SK binding upon transcriptional inhibition had no effect on the composition of Smn complexes or their recruitment of snRNAs. Instead, 7SK itself might act as a decoy RNA that blocks Smn-mediated snRNP assembly steps, such as Sm protein transfer onto snRNAs. Such a mechanism would allow cells to respond rapidly to changes in transcription by regulating the binding of 7SK to Smn, rather than disassembling and reassembling Smn complexes which would be slower and more energy-consuming. Cells might utilize such a mechanism for fine-tuning Smn complex activity in the cytosol in order to avoid a surplus of snRNPs. Excessive snRNP levels under conditions of low transcriptional demand might have detrimental effects and reduce splicing precision. Beyond that, unused snRNPs might form aggregates and impair cell viability. Neurons in particular might be vulnerable to such an imbalance between excessive snRNP availability and low splicing demand stemming from reduced transcription. In agreement with this notion, snRNP aggregates have been shown to occur in neurodegenerative disorders. For examples, aggregates of U1 snRNP have been detected in Alzheimer's disease and might contribute to the degeneration of affected neurons[39,40]. Furthermore, accumulations of snRNPs have been observed in spinal motoneurons from ALS patients[41]. Thus, dysregulated snRNP assembly might contribute to the pathogenesis of neurodegenerative disorders.

Also under physiological conditions, regulation of snRNP production might be important, particularly in development. During cellular differentiation highly active general transcription programs in undifferentiated precursor cells are gradually replaced by more specialized gene expression regulatory programs. Accordingly, the activity of the SMN complex is down-regulated during myogenic and neuronal differentiation, and during nervous system development[11]. In agreement with an inhibitory effect of 7SK binding on Smn activity, we observed an enhanced association of Smn and Larp7 in mouse brain in the same time window during which snRNP assembly was shown to decline[11]. Thus, the interaction of Smn with 7SK complexes might be part of regulatory pathways, that facilitate precise control of the activity and levels of macromolecular machineries

during terminal differentiation. This mechanism could be of particular relevance for cells that require precise control of their transcriptional output such as neurons, which utilize RNA processing mechanisms at multiple levels to maintain their extensive morphological complexity.

We found that transcriptional inhibition particularly affected the snRNA levels of U4atac and U11, and the assembly of U2 and U12 snRNP particles. This suggests that these snRNAs are more sensitive towards changes in transcription rates or towards alterations in Smn complex activity following association with the 7SK complex. Interestingly, U2, U4atac, U11, and U12 are among those snRNPs that are most strongly reduced in the spinal cord of SMA mouse models harboring low levels of Smn[42,43]. To our knowledge, the biochemical basis for the differential susceptibility of individual snRNAs to reduced SMN complex activity is presently unknown. It is possible that snRNAs differ in their affinity for the SMN complex or in their efficiency of Sm core assembly due to structural differences. Either way, the reduction of particular snRNPs that occurs upon prolonged inhibition of snRNP assembly following SMN depletion might induce splicing defects of introns, that are particularly dependent on them such as U12-dependent introns[44]. Interestingly, a role of U2 snRNA for neuronal maintenance has also been observed in mice that are homozygous for a deletion mutation in one of the U2 genes, which induced cerebellar neurodegeneration accompanied by ataxia[45]. In these mice, splicing especially of small introns was affected indicating that U2 levels are adjusted to facilitate efficient progression of particular alternative splicing events.

We found that Smn associates with 7SK through protein–protein interactions with Larp7 and Mepce. Given that Larp7 and Mepce are core components of 7SK particles it was surprising for us to observe that Smn selectively associates with 7SK/hnRNP but not with 7SK/Hexim1/P-TEFb complexes. This raises questions about the molecular basis for the selective association of Smn with 7SK/hnRNP, but not with 7SK/Hexim1/P-TEFb complexes. Previous analyses in human cell lines have shown that not only HEXIM1 binding to SL1 of 7SK, but also LARP7 binding to SL4 is required for recruitment of P-TEFb to 7SK[15]. Furthermore, a direct interaction of CDK9 with GST-tagged LARP7 has been observed[18]. In agreement, we observed co-purification of Cdk9 and Cyclin T1 with Larp7 and Mepce in the presence of RNase supporting a model in which Larp7, besides its role in 7SK stabilization, has additional roles in P-TEFb binding to 7SK through protein contacts[15]. It is therefore possible that steric hindrance prevents Smn from binding to Larp7 and Mepce in 7SK/Hexim1/P-TEFb complexes. In contrast, interactions of hnRNPs with 7SK are mediated through SL3 of 7SK and thus might allow Larp7 and Mepce to engage into additional contacts with Smn in 7SK/hnRNP complexes. We observed that, in the cytosol, hnRNP R, rather than hnRNP A1, was associated with Smn and Larp7. This suggests the possibility that separate 7SK subcomplexes associated with different RNA-binding proteins exist and that those 7SK complexes which contain hnRNP R are particularly amenable to interactions with the Smn complex.

In summary, our data indicate that the Smn complex associates with 7SK particles, and that this association is tightly linked to the transcriptional output of cells. This identifies 7SK as an important link between the dynamics of nuclear transcription and the activity of cytosolic processes such as snRNP production.

## Methods

**Animals.** CD-1 mice and *Smn+/−;SMN2* mice[38] were housed in the animal facility of the Institute of Clinical Neurobiology at the University Hospital Wuerzburg. Mice were kept under controlled conditions in a 12/12 h day/night cycle at 20–22 °C and 55–65% humidity with abundant supply of food and water.

Experiments were performed strictly following the regulations on animal protection of the German federal law and of the Association for Assessment and Accreditation of Laboratory Animal Care, in agreement with and under control of the local veterinary authority.

**Primary mouse motoneuron culture**. Spinal motoneurons were cultured as previously described[46]. Briefly, lumbar spinal cord tissues from E12.5 CD-1 mouse embryos were dissected, and motoneurons were enriched via p75NTR antibody panning. Motoneurons were cultured in neurobasal medium (Gibco) supplemented with B27 (1:50; Gibco), 2% heat-inactivated horse serum (Linaris), 500 μM GlutaMAX (Gibco), and 5 ng/ml BDNF. The medium was replaced 24 h after plating and then every second day. For co-immunoprecipitation, $1.5 \times 10^6$ motoneurons were plated in individual T-25 flasks precoated with poly-D/L-ornithine hydrobromide (Sigma, P8638) and laminin-111 (Invitrogen, 23017-015), and grown for 6 d. On day 6, motoneurons were treated with 1 μg/ml ActD or DMSO for 6 h prior to harvesting.

**NSC-34, HEK293TN, and HeLa cell culture**. NSC-34 cells (Cedarlane, cat. no. CLU140), HEK293TN cells (System Biosciences, cat. no. LV900A-1) and HeLa cells (Leibniz Institute DSMZ-German Collection of Microorganisms and Cell Cultures GmbH, DSMZ no. ACC 57) were cultured at 37 °C and 5% $CO_2$ in high glucose Dulbecco's Modified Eagle Medium (DMEM; Gibco) supplemented with 10% fetal calf serum (Linaris), 2 mM GlutaMAX (Gibco) and 1% penicillin–streptomycin (Gibco). Cells were passaged when they were 80–90% confluent. For transcriptional inhibition assays, $2 \times 10^6$ cells were grown in 10 cm dishes for 2 d. Then medium was replaced with fresh medium containing 1 μg/ml ActD treatment or DMSO as control and incubated for the indicated durations prior to harvesting.

**Immunofluorescence staining of cultured motoneurons**. For immuno-fluorescence stainings, motoneurons were cultured on 10 mm glass coverslips (Marienfeld GmbH, cat. no. 0111500) precoated with poly-D/L-ornithine hydrobromide and laminin-111 at a density of 10,000 cells per coverslip. At day 6, motoneurons were washed three times with prewarmed phosphate-buffered saline (PBS) and fixed with 4% paraformaldehyde (PFA) for 20 min at room temperature (RT), then washed three times with prewarmed PBS. For permeabilization, 0.3% Triton X-100 was applied for 20 min at RT followed by three washes with prewarmed PBS. Motoneurons were treated with 10% horse serum, 2% BSA in Tris-buffered saline with Tween 20 (TBS-T) for 0.5 h at RT to reduce unspecific binding followed by primary antibody incubation overnight at 4 °C. The cells were washed three times with TBS-T and incubated with appropriate fluorescently labeled secondary antibodies in TBS-T for 1 h at RT. Motoneurons were washed three times with TBS-T at RT, then washed once with water. Motoneurons were embedded with Aqua-Poly/Mount (Polysciences, 18606-20). The following primary and secondary antibodies were used for immunostaining: polyclonal chicken anti-GFP (Ab13970, Abcam; 1:1,000), monoclonal mouse anti-α-Tubulin antibody (T5168, Sigma; 1:1,000), polyclonal rabbit anti-Larp7 (17067-1-AP, Proteintech; 1:100), donkey anti-chicken IgG (H + L) (Alexa 488; 703-545-155, Jackson Immunoresearch; 1:800), donkey anti-mouse IgG (H + L) (Cy 3; 715-165-151, Jackson Immunoresearch; 1:800), and donkey anti-rabbit IgG (H + L) (Cy 5; 711-175-152, Jackson Immunoresearch; 1:800).

**Preparation of brain lysates**. Brains from E18 Smn+/+;SMN2 and Smn−/−; SMN2 mice[38] or E18, P2, P17, and 3 months wildtype mice were dissected and lysed in lysis buffer B [10 mM HEPES (pH 7.0), 100 mM KCl, 5 mM $MgCl_2$, 0.5% Nonidet P-40 (NP-40)]. Protein concentration was measured using a BCA kit (Thermo Fisher Scientific) and 200–500 μg total brain protein was used per co-immunoprecipitation.

**Preparation of lentiviral knockdown constructs**. shRNAs targeting Larp7, Mepce, Gemin2, and SmB/B′ were cloned into a modified version of pSIH-H1 shRNA vector (System Biosciences) containing EGFP according to the manufacturer's instructions. The following antisense sequences were used for designing shRNA oligonucleotides targeting *Larp7*: 5′-TTCTCAGCTTTGGTGATTAGC-3′, *Mepce*: 5′-TTGTTTCCGTGAGAGACTTTC-3′, *Gemin2*: 5′-ACTAAGCAGATC AGCAGATTC-3′ and *Snrpb* (encoding SmB/B′): 5′-TTGTCAAAGGCTTTGAA G-3′ (Supplementary Table 1). shRNAs for knockdown of Smn and 7SK were described before[26,47]. The plasmid for double knockdown of Mepce and Larp7 was prepared by inserting a U6 promoter followed by the shRNA targeting *Mepce* into the ClaI site of pSIH-H1 containing the shRNA against *Larp7*. Empty pSIH-H1 expressing EGFP was used as control. Lentiviral particles were packaged in HEK293T cells with pCMV-pRRE, pCMV-pRSV, and pCMV-pMD2G as described before[48]. To assess knockdown efficiency, RNA was extracted from NSC-34 cells using the NucleoSpin RNA kit (Macherey-Nagel) and reverse-transcribed with random hexamers using the First Strand cDNA Synthesis Kit (Thermo Fisher Scientific). Reverse transcription reactions were diluted 1:5 in water and cDNA levels measured by qPCR. Relative expression was calculated using the ΔΔCt method.

**Pulldown of endogenous 7SK**. NSC-34 cells were grown in 10 cm dishes until 80–90% confluency. Cells were washed once with ice-cold Dulbecco's Phosphate Buffered Saline (DPBS, without $MgCl_2$, $CaCl_2$; Sigma) and collected by scraping. Cells were lysed in 1 ml lysis buffer B [10 mM HEPES (pH 7.0), 100 mM KCl, 5 mM $MgCl_2$, 0.5% NP-40] on ice for 15 min. Lysates were centrifuged at $16,000 \times g$ for 15 min at 4 °C and the supernatant was transferred into a new tube. One hundred and fifty picomoles of biotinylated RNA oligonucleotide Biotin-scr or Biotin-pd7SK (Supplementary Table 2) were added to 600 μl NSC-34 lysate followed by rotation end-over-end for 2 h at 4 °C. Then 30 μl streptavidin magnetic beads (Pierce) were added followed by rotation for 30 min at 4 °C. Beads were washed twice with 1 ml lysis buffer B and twice with lysis buffer A [10 mM HEPES (pH 7.0), 100 mM KCl, 5 mM $MgCl_2$]. Buffer was removed completely and beads were snap-frozen in liquid $N_2$ for mass spectrometry analysis.

**Sample preparation for LC-MS/MS**. Beads were thawed and RNA was degraded by Benzonase. For this purpose 20 μl of 375 U/ml Benzonase in 50 mM Tris-HCl pH 7.5 supplemented with 2 mM $MgCl_2$ was added to each tube and incubated for 1 h. Beads were kept in solution by shaking at 750 rpm on an Eppendorf Thermomixer C at all steps. Proteins were denatured by adding 150 μl of 8 M urea 50 mM Tris-HCl pH 7.5 solution and 5 μl of 30 μM dithiothreitol and digested by LysC (0.25 μg/sample) at RT. This initial protein digestion with LysC under strongly denaturing conditions facilitates cleavage at otherwise poorly accessible sites. To permit subsequent digestion with trypsin, which is not stable at 8 M urea, samples were diluted 4-fold with 50 mM Tris-HCl pH 7.5. To alkylate cysteines, chloroacetamide was added to a final concentration of 5 mM. Trypsin (0.25 μg/ sample) was added and samples were digested at RT in the dark overnight. The digest was terminated by addition of trifluoroacetic acid (final 1% v/v) and the beads were settled by centrifugation. Half of the supernatant was further processed by desalting chromatography on three disks of C18 material using the STAGE-tip format[49]. Briefly, STAGE-tips were washed with 100 μl buffer B (50% v/v acetonitrile, 0.5% v/v acetic acid), conditioned with 100 μl methanol, washed twice with 100 μl buffer A (2% v/v acetonitrile, 0.5% v/v acetic acid), loaded with sample peptides, washed twice with 100 μl buffer A, and subjected to peptide elution by 60 μl of buffer B. The eluate was evaporated to dryness in a vacuum concentrator. Peptides were resuspended in 10 μl 2% v/v acetonitrile, 0.5% v/v acetic acid, 0.1% v/v trifluoroacetic acid, and stored at −20 °C and 2 μl were later used for mass spectrometry.

**LC-MS/MS**. Peptides were separated on an EASY-nLC 1000 HPLC system (Thermo Fisher Scientific) via in-house packed columns (75 μm inner diameter, 50 cm length, and 1.9 μm C18 particles [Dr. Maisch GmbH]) in a gradient of buffer A (0.5% formic acid) to buffer B (80% acetonitrile, 0.5% formic acid). The gradient started at 5% B, increasing to 30% B in 65 min, further to 95% B in 10 min, staying at 95% B for 5 min, decreasing to 5% B in 5 min, and staying at 5% B for 5 min at a flow rate of 300 nl/min and a temperature of 60 °C. A Quadrupole Orbitrap mass spectrometer (Q Exactive HF-X; Thermo Fisher Scientific) was directly coupled to the LC via a nano-electrospray source. The Q Exactive HF-x was operated in a data-dependent mode. The survey scan range was set from 300 to 1650 $m/z$, with a resolution of 60,000 at $m/z$ 200. Up to the 12 most abundant isotope patterns with a charge of two to five were isolated and subjected to collision-induced dissociation fragmentation at a normalized collision energy of 27, an isolation window of 1.4 Th, and a MS/MS resolution of 15,000 at $m/z$ 200. Dynamic exclusion to minimize resequencing was set to 30 s.

**Mass spectrometry data processing and bioinformatic analysis**. To process MS raw files, we employed the MaxQuant software version 1.6.0.15[50], searching against the UniProtKB mouse FASTA database using canonical and isoform protein sequences. Default search parameters were utilized unless stated differently. A false discovery rate (FDR) cutoff of 1% was applied at the peptide and protein level. The search feature "Match between runs", which allows the transfer of peptide identifications in the absence of sequencing after nonlinear retention time alignment, was enabled with a maximum retention time window of 0.7 min. Protein abundances were normalized with the MaxLFQ label-free normalization algorithm[51]. For bioinformatic analysis and visualization, we used the open PERSEUS environment version 1.5.2.11[52]. Common contaminants and peptides only identified by site modification were excluded from further analysis. Protein abundance values were log₂-transformed. Data were filtered for proteins with at least three valid values (experimental observations) in the four replicates in at least one group (Biotin-pd7SK or Biotin-scr). Missing values were imputed from a normal distribution using the default parameters (width = 0.3 and down-shift = 1.8 standard deviations of parent distribution). Based on principal component analysis, the fourth of four replicates of both the Biotin-pd7SK and Biotin-scr interaction proteomes were outliers and thus removed from further analysis. The Volcano plot was generated with the built-in PERSEUS tool using $t$-test statistics combined with a permutation-based FDR of 5%[53], and an s0-parameter of 0.1 for integrating the fold change effect size. Proteins above the cutoff line have a $q$-value below 5%. The 1D annotation enrichment analysis was performed with the built-in PERSEUS tool using a two-sample Wilcoxon–Mann–Whitney test on the difference (Biotin-pd7SK vs. Biotin-scr) of mean log₁₀-transformed intensities[54]. Annotation terms of

the following categories were included in the analysis: gene ontology (GO) molecular function (GOMF), biological process (GOBP), cellular compartment (GOCC), UniProt Keywords, UniProt protein families, and UniProt "Interacts with" annotation. Moreover, a custom annotation for Mus Musculus RNA-binding proteins based on available data from RBPDB (http://rbpdb.ccbr.utoronto.ca/proteins.php?species_filter=10090)[55] was included. Of the listed 415 proteins, 216 proteins were contained in the interaction proteomics dataset. In the enrichment analysis, positive enrichment scores correspond to higher intensities in Biotin-pd7SK samples, whereas negative enrichment scores correspond to higher intensities in Biotin-scr samples. For visualization, the logit value of the enrichment score was plotted against the $-\log_{10}$-transformed Benjamini–Hochberg-adjusted P-values.

**Co-immunoprecipitation.** Cells were washed once with ice-cold DPBS and collected by scraping. Cells were lysed in 1 ml lysis buffer B [10 mM HEPES (pH 7.0), 100 mM KCl, 5 mM $MgCl_2$, 0.5% NP-40] on ice for 15 min. Lysates were centrifuged at $20,000 \times g$ for 15 min at 4 °C. Ten microliter of Dynabeads Protein G or A (depending on the antibody species) (Invitrogen) and 1 μg antibody or IgG control were added to 200 μl lysis buffer and rotated for 30–40 min at RT. Then 200 μl lysate were added to the antibody-bound beads and rotated for 2 h at 4 °C. Beads were washed twice with lysis buffer B and proteins were eluted in 1× Laemmli buffer [50 mM Tris-HCl (pH 6.8), 1% sodium dodecyl sulfate, 6% glycerol, 1% β-mercaptoethanol, 0.004% bromophenolblue]. Proteins were size-separated by SDS-PAGE and analyzed by immunoblotting. For RNase treatment, 3 μl RNase A (Thermo Fisher Scientific) were added to 200 μl lysate and incubated at 37 °C for 10 min before proceeding with co-immunoprecipitation.

**Sucrose gradient centrifugation.** NSC-34 cells were lysed in lysis buffer B [10 mM HEPES (pH 7.0), 100 mM KCl, 5 mM $MgCl_2$, 0.5% NP-40] and protein concentration was measured by Bradford assay. Lysates were diluted to a final concentration of 1.5 μg/μl and 200 μl were layered onto a 5–20% sucrose gradient. Gradients were centrifuged at $225,000 \times g$ for 3 h at 4 °C followed by fractionation. The same volume of each fraction was used for Western blotting or qPCR.

**RNA immunoprecipitation.** NSC-34 cells were lysed in 1 ml lysis buffer B [10 mM HEPES (pH 7.0), 100 mM KCl, 5 mM $MgCl_2$, 0.5% NP-40] for 15 min on ice and then centrifuged for 15 min at $20,000 \times g$. One microgram of antibody or control IgG was bound to 10 μl Protein A or G Dynabeads (depending on the antibody species) in 200 μl lysis buffer B at RT for 30-40 min. Hundred microliter of lysate was used as input sample, and 400 μl lysate was incubated with the antibody-bound beads for 2 h at 4 °C by rotation. Beads were washed twice with 400 μl lysis buffer B. RNA was extracted from the input sample with 1 ml TRIzol (Thermo Fisher Scientific) and from the beads with 200 μl TRIzol. RNA was reverse-transcribed with random hexamers using the First Strand cDNA Synthesis kit (Thermo Fisher Scientific). Reverse transcription reactions were diluted 1:5 in water and transcript levels were measured by qPCR. Relative RNA binding was calculated using the ΔΔCt method with normalization to input levels.

**In vitro snRNP assembly assay.** The snRNP assembly assay was performed as described before[32,33] with the following modifications. For preparation of biotinylated U2 snRNA, the mouse U2 snRNA gene was PCR-amplified with primers T7-U2-F and U2-R (Supplementary Table 3). In vitro transcription was performed with T7 RNA Polymerase (Thermo Fisher Scientific) with a 1:10 ratio of biotin-16-UTP/UTP (Jena Bioscience, NU-821-BIO16) and 0.25 U/50 μl inorganic pyrophosphatase (Thermo Fisher Scientific) overnight at 37 °C. Afterwards, the DNA template was digested by addition of 4 U of TURBO DNase (Ambion) and incubation for 30 min at 37 °C. RNA was then purified on a NucleoSpin RNA column (Macherey-Nagel) and an aliquot was subjected to agarose gel electrophoresis for size confirmation.

Cytosolic extracts were obtained as following. NSC-34 cells were grown in 10 cm dishes until 80–90% confluency. Cells were treated with 1 μg/ml ActD or with DMSO for 6 h, washed once with ice-cold DPBS and collected by scraping. Cells were transferred into 15 ml Falcon tubes and collected by centrifugation at $300 \times g$ for 0.5 min at 4 °C. Cells were lysed in one volume of lysis buffer A [10 mM HEPES (pH 7.0), 100 mM KCl, 5 mM $MgCl_2$] containing 50 μg/ml digitonin (Calbiochem) followed by centrifugation at $1700 \times g$ for 1 min. Supernatants containing cytosolic extract were collected and NP-40 was added to a final concentration of 0.01%. Following further centrifugation at $10,600 \times g$ for 15 min at 4 °C, supernatants were obtained and stored on ice. Two microliter of extract were used for measuring protein concentration with the BCA kit.

The assembly assay was conducted as following. For each reaction, 200 ng biotin-labeled U2 snRNA were dissolved in 5 μl water and heated at 75 °C for 5 min followed by incubation on ice for 3 min. After adding 0.56 μl 10× lysis buffer A, the RNA was incubated at 37 °C for 5 min and then kept on ice. The RNA was added to 50 μl NSC-34 cytosolic extract containing 100 μg total protein and reactions were carried out at 30 °C for 15 min with or without 2.5 mM ATP. Meanwhile, 10 μl streptavidin magnetic beads were prewashed once with lysis buffer A containing 0.01% NP-40 and then dissolved in 200 μl of the same buffer and kept

on ice. After 15 min, the reaction mix was added to the streptavidin beads and the tube was rotated at 4 °C for 20 min. Beads were washed twice with wash buffer [10 mM HEPES (pH 7.0), 500 mM KCl, 5 mM $MgCl_2$, 0.01% NP40]. Proteins were eluted in 1× Laemmli buffer and analyzed by Western blotting.

**RNA pulldown assay.** 7SK RNAs were generated by in vitro transcription as described before[26] and were purified using the NucleoSpin RNA kit (Macherey-Nagel). Then 2 μg 7SK-WT, 2 μg 7SK-AS, 1.42 μg 7SK-ΔSL1, 1.65 μg 7SK-ΔSL2, 1.57 μg 7SK-ΔSL3s, 1.4 μg 7SK-ΔSL3l, 0.435 μg 7SK-ΔSL123, or 1.5 μg ΔSL123-GFP were dissolved in 20 μl water and heated to 65 °C for 5 min followed by incubation on ice for 3 min. Twenty-two picomoles biotinylated DNA oligonucleotide Biotin-pd7SK-as or Biotin-pd7SK-s (Supplementary Table 2) were added to the RNAs and incubated in a total volume of 50 μl lysis buffer B [10 mM HEPES (pH 7.0), 100 mM KCl, 5 mM $MgCl_2$, 0.5% NP-40] for 10 min at 37 °C and then kept on ice. To test RNA binding to streptavidin beads, 200 μl lysis buffer B and 10 μl streptavidin magnetic beads were added to the 7SK RNA/biotin oligonucleotide mixtures and incubated for 30 min at 4 °C. Beads were washed twice with 400 μl lysis buffer B. RNA was released by adding RNA elution buffer (95% formamide, 10 mM EDTA, 0.025% bromophenol blue) and heating at 70 °C for 10 min. For visualization, RNA was subjected to agarose gel electrophoresis on a 2.5% Tris-borate-EDTA agarose gel. To pull down proteins, 200 μl NSC-34 lysate was added to the 7SK RNA/biotin oligonucleotide mixture and incubated for 2 h at 4 °C. Then 10 μl of streptavidin beads were added followed by incubation at 4 °C for 30 min. Beads were washed twice with 400 μl lysis buffer B, and proteins were eluted in 1× Laemmli buffer for Western blot analysis.

**Subcellular fractionation.** $2 \times 10^6$ NSC-34 cells were grown in 10 cm dishes for 2 d. Cells were washed once with ice-cold DPBS and fractionated in three steps. Step 1: 4 ml lysis buffer A [10 mM HEPES (pH 7.0), 100 mM KCl, 5 mM $MgCl_2$] containing 35 μg/ml digitonin were added per dish and incubated for 10 min at 4 °C. The supernatant was transferred from the dish to a 15 ml Falcon tube and kept on ice. Step 2: The cells remaining on the dish were washed once with ice-cold DPBS and lysed in 4 ml lysis buffer B [10 mM HEPES (pH 7.0), 100 mM KCl, 5 mM $MgCl_2$, 0.5% NP-40]. The lysate was transferred into 15 ml Falcon tubes and kept on ice for 15 min. Meanwhile, the lysate from Step 1 was centrifuged at $2000 \times g$ for 5 min at 4 °C and the supernatant containing the cytosolic fraction (Cyt) was transferred to a new tube. The lysate from Step 2 was centrifuged at $20,000 \times g$ for 15 min at 4 °C and the supernatant containing nuclear soluble and organellar proteins (Nuc) was transferred into a new tube. Step 3: The pellet from centrifugation of the Nuc fraction was washed once with ice-cold DPBS, dissolved in 4 ml lysis buffer B and sonicated. This fraction (Chr) contains nuclear insoluble proteins. The same volume of each fraction was used for Western blotting and qPCR. For co-immunoprecipitation, 800 μl of each fraction was used.

For fractionation of motoneurons, 400,000 motoneurons were plated per well in 24 well plates and cultured for 7 d. Cells were washed once with ice-cold DPBS and fractionated in three steps. Step 1: 300 μl lysis buffer A containing 200 μg/ml digitonin were added per well and incubated for 10 min at 4 °C. The supernatant was transferred from the well to a 1.5 ml tube and kept on ice. Step 2: The cells remaining on the well were washed once with ice-cold DPBS and lysed in 300 μl lysis buffer B. The lysate was transferred into a 1.5 ml tube and kept on ice for 15 min. Meanwhile, the lysate from Step 1 was centrifuged at $2000 \times g$ for 5 min at 4 °C and the supernatant containing the Cyt fraction was transferred to a new tube. The lysate from Step 2 was centrifuged at $20,000 \times g$ for 15 min at 4 °C and the supernatant containing the Nuc fraction was transferred into a new tube. Step 3: The pellet from centrifugation of the Nuc fraction was washed once with ice-cold DPBS, dissolved in 300 μl lysis buffer B and sonicated to obtain the Chr fraction. The same volume of each fraction was used for Western blotting and qPCR.

**qPCR.** Primers are listed in Supplementary Table 4. Primers for qPCR of snRNAs were from Lotti et al.[56]. Reactions were set up with Luminaris HiGreen qPCR Master Mix (Thermo Fisher Scientific) on a LightCycler® 96 (Roche).

**Western blot quantification.** The intensity of the protein bands was quantified using Fiji[57].

**Antibodies.** Antibodies used throughout the study are listed in Supplementary Table 5.

**Statistics and reproducibility.** Statistical analyses were performed using Graph-Pad Prism version 6 for Windows (GraphPad Software, San Diego, California USA). All results shown are representative of at least three independent experiments.

**Reporting summary**. Further information on research design is available in the Nature Research Reporting Summary linked to this article.

## Data availability

The mass spectrometry proteomics data have been deposited to the ProteomeXchange Consortium via the PRIDE[58] partner repository with the dataset identifier PXD021360. Figure 1a and Supplementary Fig. 1d, e have associated raw data in Supplementary Data 1–3. All data is available from the corresponding author upon reasonable request. Source data are provided with this paper.

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

## Acknowledgements

This work was supported by the Deutsche Forschungsgemeinschaft [BR4910/1–1 (SPP1738) and BR4910/2–1 (SPP1935) to M.B.; SE697/4–1 (SPP1738), SE697/5–1 (SPP1935) and SE697/1 to M.S.; JA1823/3-1 to S.J.; Fi573/15-2 (SPP1935) and Fi573/20-1 to U.F.] and the European Research Council Synergy Grant under FP7 GA number ERC-2012-SyG_318987-Toxic Protein Aggregation in Neurodegeneration (ToPAG) to J.B., F.M., and M.M. This publication was supported by the Open Access Publication Fund of the University of Wuerzburg.

## Author contributions

Conceptualization, C.J., M.S., and M.B.; Methodology, C.J., M.S., and M.B.; Software, J.B.; Formal analysis, C.J., J.B., and M.B.; Investigation, C.J., J.B., P.R., and L.H.; Resources, F.M., S.J., M.M., U.F., and M.S.; Writing—Original draft, C.J., M.S., and M.B.; Writing—Review and editing, C.J., M.S., and M.B.; Visualization, C.J., J.B., and M.B.; Supervision, F.M., S.J., U.F., M.S., and M.B.; Funding acquisition, M.S. and M.B.

## Competing interests

The authors declare no competing interests.
