## [Peer Review File · Nature Communications]

REVIEWER COMMENTS

Reviewer #1 (Remarks to the Author):

Review of Ji et al Nat Comms

Altogether, I would say this is a rather good manuscript. It ties together a number of loose threads from the older literature and uncovers several new interactions that help to explain how the transcriptional apparatus communicates with the cytoplasmic RNP assembly machinery. The experimental outline is logical and procedures are well performed. Most of my critiques are minor, as detailed below.

The one thing I think the authors should address (by way of experiment or explanation) has to do with fleshing out the components that are present in this 'new' combined 7SK/hnRNP/SMN particle.

1. As referenced by the authors (#24), in 2007 Tamas Kiss' lab showed that 7SK forms what they termed a '7SK/hnRNP' complex. Levels of this second complex are increased during transcriptional inhibition. Their complex included hnRNPs-R, -A1, and -Q, along with RNA helicase A (RHA), and is almost certainly the same one described here.

2. Also many years ago, Dreyfuss and colleagues (Pellizzoni 2001, JCB) showed that SMN interacts with RHA in a mass spec screen and identified a complex in co-IPs. Although there are several things about that paper that are likely incorrect, I wonder if this is part of the same complex.

Therefore, I would like to know if the authors have detected RHA in any of their 7SK pulldowns (either by western blotting or by mass spectrometry)? Furthermore, what about other SMN associated proteins? Does the 7SK/hnRNP/SMN particle contain Gemin3/4/5 or Gemin6/7/8? No need to blot for all of them, but at least comment on presence/absence of the other Gemin.

Minor points:

a. The title is a bit awkward. Now that I have read the paper I understand what they mean, but at first glance it seems obvious and trivial. Of course, transcription regulates snRNP production. For better effect, the authors should rephrase it. Turn sentence around or qualify the context of the word 'transcription.'

b. Line 103. Qualify this statement as well. Pol II does not tend to pause at every gene. So maybe: "In a large subset of transcripts, ..."

c. Line 115: maybe move ref 24 to the end of the previous sentence on line 114. Otherwise the statement on 114 needs some support.

d. Line 144: Is this localization pattern unique to MNs or is it also found in other/all cell types? Not clear from this paragraph if this is a general phenomenon or if it may be specific for neuromuscular development.

e. Line 178: see above, what about other Gemin components?

f. Line 226: replace 'balance between' with 'composition of'?

g. Line 258: ActD would certainly intercalate in DNA that encodes snRNA genes also. So isn't it obvious that snRNA transcription would go down under these circumstances?

h. Line 260: In support of the fact that U4atac and U11 showed strongest decrease following ActD, both Gabanella (2007) and Praveen et al (2012) showed that loss of SMN affects these snRNAs first and that this is a conserved feature of metazoan SMN systems.

i. Lines 334-35: Might not these 'other' RNAs help mediate interaction between SMN and hnRNP-R when Larp7 and 7SK are not reduced? What is the evidence that would suggest this happens only under depletion conditions?

j. Figures 4 (and 6): add or change labels for fractions F1, F2 and F3 to be something more descriptive so readers can know what they are at a glance.

k. Lines 451-453. Too many thoughts crammed into one sentence (esp the last phrase after comma).

Break up?

Reviewer #2 (Remarks to the Author):

In this work Ji and coworkers describe a novel function for the 7SK complex in spliceosomal snRNP biogenesis. The authors perform a lengthy and detailed biochemical characterization of the 7SK snRNP in association with hnRNPs and component of the SMN complex. The hnRNPA1/R 7SK complex appears to be present both in the nuclei and in the cytoplasm and to bind Smn and Gemin2, key component of the SMN complex but not HEXIM and P-TEFb component of the classic 7SK particle, which has well defined functions in transcription. Interestingly, the assembly of the 7SK/HEXIM/P-TEFb and the 7SK/hnRNPA1-R/SMN complexes are both regulated by transcription inhibition although with opposite outcomes. While inhibition of transcription decreases the assembly of the 7SK/HEXIM/P-TEFb it also increases the 7SK/hnRNPA1-R/SMN complex, which results in a decrease in the assembly of some mature spliceosomal snRNP.

Overall this work is well executed, with proper scientific rigor and controls and the majority of the author's claims are well supported by the results presented.

Nevertheless, some results should be better clarified and discussed:

In extended figure 5C hnRNPA1 seems not to be present in any of the IP carried out, since SmB/B' is also associated with 7SK, shouldn't A1 be present as well?

The results presented in extended figure 5 also suggest that RNAs other than 7SK might be also responsible for the assembly of a SMN/LARP7/hnRNPR (possibly A1) containing complex, what do we know of LARP7 RNA binding specificity? What RNAs are purified by LARP7 pulldown?

It is unclear why LARP7 MEPCE 7SK complex would interfere with the assembly of some snRNP (U2) but not others (U1)? This should be discussed more in detail.

Reviewer #3 (Remarks to the Author):

In the present article entitled "Transcription regulates snRNP production through interaction of 7SK with the Smn complex" Changhe Ji et al. investigate the association of Smn complex with 7SK complex involved in transcriptional regulation providing important information about how transcriptional control is linked to the spliceosome and biogenesis. The manuscript is well done, writing is concise and simple and figures are very complete and illustrative. Before its publication in Nature Communications I would suggest doing these little changes:

Figures:

Fig 1a. Maybe it could be illustrative to highlight in the volcano plot the protein Larp7 too as authors mention it in the text.

Discussion:

Line 459. Remove the question.

Materials and Methods:

Line 660. Clarify briefly the reason to use LysC for digestion prior trypsin.

Data in PRIDE: is the data already published in PRIDE? Otherwise it is good to provide username and password for reviewers in the manuscript.

REVIEWER COMMENTS

Reviewer #1 (Remarks to the Author):

Review of Ji et al Nat Comms

Altogether, I would say this is a rather good manuscript. It ties together a number of loose threads from the older literature and uncovers several new interactions that help to explain how the transcriptional apparatus communicates with the cytoplasmic RNP assembly machinery. The experimental outline is logical and procedures are well performed. Most of my critiques are minor, as detailed below.

The one thing I think the authors should address (by way of experiment or explanation) has to do with fleshing out the components that are present in this 'new' combined 7SK/hnRNP/SMN particle.

1. As referenced by the authors (#24), in 2007 Tamas Kiss' lab showed that 7SK forms what they termed a '7SK/hnRNP' complex. Levels of this second complex are increased during transcriptional inhibition. Their complex included hnRNPs-R, -A1, and -Q, along with RNA helicase A (RHA), and is almost certainly the same one described here.

2. Also many years ago, Dreyfuss and colleagues (Pellizzoni 2001, JCB) showed that SMN interacts with RHA in a mass spec screen and identified a complex in co-IPs. Although there are several things about that paper that are likely incorrect, I wonder if this is part of the same complex.

Therefore, I would like to know if the authors have detected RHA in any of their 7SK pulldowns (either by western blotting or by mass spectrometry)?

Author's response: We agree with the reviewer that in addition to hnRNP R and A1, both of which we validated as 7SK interactors in our manuscript, other components of 7SK/hnRNP particles such as hnRNP A2/B1, Q, and RHA, which were detected by van Herreweghe et al. in 2007 (DOI: 10.1038/sj.emboj.7601783), might be components of the 7SK particles that we purified. We have followed up on the suggestion by the reviewer in two ways. First, we included an additional volcano plot of the proteomics data in Supplementary Fig. 1, in which hnRNP A1, A2/B1, Q, R and RHA were labeled. This showed that these proteins except RHA were co-purified by 7SK pulldown together with the Smn complex. It is possible that the 7SK oligonucleotide we used for 7SK pulldown interfered with RHA binding to 7SK, which might explain why we did not detect it in our proteomics data. Therefore, as a second approach, we conducted additional immunoprecipitations of MePCE and LARP7 from NSC-34, HEK293TN and HeLa cells. With both antibodies we were able to detect co-purification of hnRNP Q and RHA in all three cell lines. Additionally, we observed co-immunoprecipitation of RHA with Smn in NSC-34, HeLa and HEK293TN cells in agreement with a previous study mentioned by the reviewer (Pellizzoni et al. 2001 JCB DOI: 10.1083/jcb.152.1.75). We now included these new data in Supplementary Fig. 2. In the Results section we added the following:

"In addition to hnRNP R and hnRNP A1, other RNA-binding proteins have previously been identified as 7SK interactors including hnRNP Q and RHA^{24,25}. In line with these studies, we observed co-purification of hnRNP Q and RHA with anti-LARP7 and anti-MePCE (Supplementary Fig. 2g-l). SMN has previously been shown to interact with RHA³⁰. In

agreement, we found that RHA co-immunoprecipitated with Smn in NSC-34, HeLa and HEK293TN cells (Supplementary Fig. 2p-r). Taken together, our results indicate that Smn selectively associates with 7SK complexes containing hnRNP A1 and R, and that hnRNP Q and RHA are also part of complexes involving 7SK and Smn.”

Furthermore, what about other SMN associated proteins? Does the 7SK/hnRNP/SMN particle contain Gemin3/4/5 or Gemin6/7/8? No need to blot for all of them, but at least comment on presence/absence of the other Gemins.

Author’s response: We thank the reviewer for this suggestion. In the proteomics data we observed co-purification of Gemin2-5 by 7SK pulldown (Fig. 1a). We have immunoprecipitated LARP7 and MePCE from NSC-34, HEK293TN and HeLa cells and observed co-purification of GEMIN3 and 4. Given that SMN stably associates with GEMIN2, and GEMIN3 with GEMIN4 and 5 (Battle et al. 2007 JBC DOI: 10.1074/jbc.M702317200), this suggests that 7SK particles associate with SMN/GEMIN2 and GEMIN3/4/5 subcomplexes. We included these new data in Supplementary Fig. 2.

Minor points:

a. The title is a bit awkward. Now that I have read the paper I understand what they mean, but at first glance it seems obvious and trivial. Of course, transcription regulates snRNP production. For better effect, the authors should rephrase it. Turn sentence around or qualify the context of the word ‘transcription.’

Author’s response: We modified the title accordingly. It now is:

“Interaction of 7SK with the Smn complex modulates snRNP production”.

b. Line 103. Qualify this statement as well. Pol II does not tend to pause at every gene. So maybe: “In a large subset of transcripts, ...”

Author’s response: We thank the reviewer for this suggestion and have modified the sentence accordingly. It now is:

“For a large subset of genes, RNA polymerase II tends to pause downstream of the transcription initiation site shortly after transcription initiation¹².”

c. Line 115: maybe move ref 24 to the end of the previous sentence on line 114. Otherwise the statement on 114 needs some support.

Author’s response: We combined both sentences because reference 24 (Van Herreweghe et al. 2007 EMBO J DOI: 10.1038/sj.emboj.7601783) supports both statements. The new sentence is:

“The 7SK/hnRNP complexes are separate from 7SK/HEXIM1/P-TEFb complexes and the balance between these 7SK subcomplexes is determined by the transcriptional activity of a cell²⁴.”

d. Line 144: Is this localization pattern unique to MNs or is it also found in other/all cell types? Not clear from this paragraph if this is a general phenomenon or if it may be specific for neuromuscular development.

Author's response: We previously detected 7SK/hnRNP R complexes in cytoplasmic fractions of motoneurons and NSC-34 cells (Briese et al. 2018 PNAS DOI: 10.1073/pnas.1721670115). Therefore, we re-phrased the sentence as following:

“Based on previous observations that 7SK/hnRNP R complexes are present in cytoplasmic fractions of NSC-34 cells and motoneurons²⁶, we searched for new protein interactors of this non-coding RNA in NSC-34 cells, a cell line with motoneuron-like properties²⁸.”

In the revised manuscript we have now included data showing that 7SK/SMN complexes exist also in HEK293TN and HeLa cells (Supplementary Fig. 2). Thus, given that SMN is abundant in the cytosol, it can be assumed that such cytoplasmic complexes also exist in other cell types beyond motoneurons.

e. Line 178: see above, what about other Gemin components?

Author's response: We have performed additional LARP7 and MePCE immunoprecipitations in NSC-34, HEK293TN and HeLa cells and detected co-purification of GEMIN3 and 4. We included these data in Supplementary Fig. 2. In the proteomics data we did not observe enrichment of Gemin6, 7 or 8 by 7SK pulldown indicating that these Gemin, which are known to form a stable subcomplex (Battle et al. 2007 JBC DOI: 10.1074/jbc.M702317200), are not associated with 7SK particles.

f. Line 226: replace 'balance between' with 'composition of'?

Author's response: We have replaced 'balance between' with 'composition of'.

g. Line 258: ActD would certainly intercalate in DNA that encodes snRNA genes also. So isn't it obvious that snRNA transcription would go down under these circumstances?

Author's response: We agree with the reviewer that, as expected, ActD acts as a transcriptional inhibitor also on snRNA genes. We now rephrased the section as following:

“Next, we analyzed total levels of Sm-class snRNAs by qPCR following ActD exposure (Fig. 3e). In line with the activity of ActD as a transcriptional inhibitor we observed reduced snRNA levels upon ActD treatment relative to DMSO control. Nevertheless, we found that the amounts of individual snRNAs were differentially affected by transcriptional inhibition.”

h. Line 260: In support of the fact that U4atac and U11 showed strongest decrease following ActD, both Gabanella (2007) and Praveen et al (2012) showed that loss of SMN affects these snRNAs first and that this is a conserved feature of metazoan SMN systems.

Author's response: We thank the reviewer for pointing out the similarities between our study and previous studies on SMA models in which SMN levels are depleted. Indeed, we found that the levels of U4atac and U11 snRNAs, and the assembly of U2 and U12 snRNPs are particularly reduced by transcriptional inhibition. These snRNPs are those that are particularly affected in the spinal cord of SMA mouse models (Gabanella et al. 2007 Plos One DOI: 10.1371/journal.pone.0000921, Zhang et al. 2008 Cell DOI: 10.1016/j.cell.2008.03.031). Thus, snRNPs might differ with respect to their sensitivity to reductions in Smn activity. We have mentioned these SMA studies in the Discussion of our manuscript as following:

“We found that transcriptional inhibition particularly affected the snRNA levels of U4atac and U11, and the assembly of U2 and U12 snRNP particles. This suggests that these snRNAs are more sensitive towards changes in transcription rates or towards alterations in Smn complex activity following association with the 7SK complex. Interestingly, U2, U4atac, U11 and U12 are among those snRNPs that are most strongly reduced in the spinal cord of SMA mouse models harboring low levels of Smn^{42,43}. To our knowledge, the biochemical basis for the differential susceptibility of individual snRNAs to reduced SMN complex activity is presently unknown. It is possible that snRNAs differ in their affinity for the SMN complex or in their efficiency of Sm core assembly due to structural differences. Either way, the reduction of particular snRNPs that occurs upon prolonged inhibition of snRNP assembly following SMN depletion might induce splicing defects of introns that are particularly dependent on them such as U12-dependent introns⁴⁴.”

42. Zhang, Z. et al. SMN deficiency causes tissue-specific perturbations in the repertoire of snRNAs and widespread defects in splicing. *Cell* 133, 585-600 (2008).

43. Gabanella, F. et al. Ribonucleoprotein assembly defects correlate with spinal muscular atrophy severity and preferentially affect a subset of spliceosomal snRNPs. *PLoS. One.* 2, e921 (2007).

44. Doktor, T. K. et al. RNA-sequencing of a mouse-model of spinal muscular atrophy reveals tissue-wide changes in splicing of U12-dependent introns. *Nucleic Acids Res.* 45, 395-416 (2017).

i. Lines 334-35: Might not these ‘other’ RNAs help mediate interaction between SMN and hnRNP-R when Larp7 and 7SK are not reduced? What is the evidence that would suggest this happens only under depletion conditions?

Author’s response: We agree with the reviewer that additional RNAs might help to stabilize 7SK/hnRNP R/Smn complexes and we investigated this possibility further. Such a candidate RNA would be β -actin mRNA, which has previously been shown to interact with hnRNP R in a Smn-dependent manner (Rossoll et al. 2003 *J Cell Biol* DOI: 10.1083/jcb.200304128). We immunoprecipitated Larp7 from NSC-34 cells and observed that high levels of β -actin mRNA were co-purified alongside 7SK. We also found that U6 snRNA was co-purified at lower levels, which is in agreement with previous studies (Muniz et al. 2013 *NAR* DOI: 10.1093/nar/gkt159, Krueger et al. 2008 *NAR* DOI: 10.1093/nar/gkn061) and supports the specificity of the immunoprecipitation. Importantly, we observed an analogous result, i.e. co-purification of high amounts of β -actin mRNA and lower amounts of U6 snRNA, when we immunoprecipitated Mepce, another component of core 7SK RNPs. Our data thus suggest that 7SK particles are associated with β -actin mRNA, which might contribute to the stability of 7SK/hnRNP R/Smn complexes. We added these new data as Supplementary Fig. 5e and f. It is possible that depletion of 7SK or Larp7 differentially modifies the association of hnRNP R with β -actin mRNA and Smn, which could be dissected in more detail in future studies using 7SK and Larp7 knockout cells and mouse models.

j. Figures 4 (and 6): add or change labels for fractions F1, F2 and F3 to be something more descriptive so readers can know what they are at a glance.

Author’s response: We have followed the reviewer’s suggestion and changed F1, F2 and F3 to Cyt, Nuc and Chr, respectively.

k. Lines 451-453. Too many thoughts crammed into one sentence (esp the last phrase after comma). Break up?

Author's response: We agree with the reviewer and have split this sentence into two as following:

“We report that Smn interacts with the 7SK core components Larp7 and Mepce. This interaction is enhanced upon transcriptional inhibition, leading to reduced snRNP levels.”

Reviewer #2 (Remarks to the Author):

In this work Ji and coworkers describe a novel function for the 7SK complex in spliceosomal snRNP biogenesis. The authors perform a lengthy and detailed biochemical characterization of the 7SK snRNP in association with hnRNPs and component of the SMN complex. The hnRNPA1/R 7SK complex appears to be present both in the nuclei and in the cytoplasm and to bind Smn and Gemin2, key component of the SMN complex but not HEXIM and P-TEFb component of the classic 7SK particle, which has well defined functions in transcription. Interestingly, the assembly of the 7SK/HEXIM/P-TEFb and the 7SK/hnRNPA1-R/SMN complexes are both regulated by transcription inhibition although with opposite outcomes. While inhibition of transcription decreases the assembly of the 7SK/HEXIM/P-TEFb it also increases the 7SK/hnRNPA1-R/SMN complex, which results in a decrease in the assembly of some mature spliceosomal snRNP.

Overall this work is well executed, with proper scientific rigor and controls and the majority of the author's claims are well supported by the results presented. Nevertheless, some results should be better clarified and discussed:

In extended figure 5C hnRNPA1 seems not to be present in any of the IP carried out, since SmB/B' is also associated with 7SK, shouldn't A1 be present as well?

Author's response: The reviewer correctly observed that we did not detect an association of hnRNP A1 with SmB/B' under control or under knockdown conditions in Supplementary Fig. 5c. Conspicuously, hnRNP A1 was also not co-purified with SmB/B' from mouse brains (Fig. 5f). We reasoned that, compared to hnRNP R, the association of hnRNP A1 with SmB/B' is less stable. While both hnRNP R and hnRNP A1 are associated with 7SK, it is possible that they form individual 7SK/hnRNP A1 and 7SK/hnRNP R subcomplexes that differ with respect to their affinity for SmB/B'. In support of this notion, hnRNP A1 and R distributed differently across fractions following sucrose gradient ultracentrifugation with hnRNP A1 being present mostly in upper fractions while hnRNP R co-sedimented with Smn in lower fractions (Fig. 1e). In our revised manuscript we now added the following sentences to the Results section:

"We did not observe an association of hnRNP A1 with SmB/B' under the control and knockdown conditions suggesting that the interaction of SmB/B' with hnRNP A1 is less stable compared to that with hnRNP R. While both hnRNP R and hnRNP A1 are associated with 7SK, it is possible that they form individual 7SK/hnRNP A1 and 7SK/hnRNP R subcomplexes that differ with respect to their affinity for SmB/B'."

and:

"Of note, while we detected co-precipitation of hnRNP R with SmB/B' we did not observe an association of hnRNP A1 with SmB/B' in mouse brains. This agrees with our previous result in NSC-34 cells (Supplementary Fig. 5c) and further indicates that hnRNP A1 is less stably associated with SmB/B'."

The results presented in extended figure 5 also suggest that RNAs other than 7SK might be also responsible for the assembly of a SMN/LARP7/hnRNPR (possibly A1) containing complex, what do we know of LARP7 RNA binding specificity? What RNAs are purified by LARP7 pulldown?

Author's response: We agree with the reviewer that, in addition to 7SK, other RNAs might contribute to the formation of Smn/Larp7/hnRNP R complexes. To provide evidence for this possibility we investigated the association of Larp7 with β -actin mRNA, a known major target of hnRNP R and Smn (Rossoll et al. 2003 JCB DOI: 10.1083/jcb.200304128, Briese et al. 2018 PNAS DOI: 10.1073/pnas.1721670115), by RNA immunoprecipitation. We found that Larp7 strongly interacts with β -actin mRNA, similar to 7SK. To validate the specificity of this interaction we also measured Larp7 association with U6 snRNA, which has previously been demonstrated to interact with Larp7, albeit at lower efficiency (Muniz et al. 2013 NAR DOI: 10.1093/nar/gkt159, Krueger et al. 2008 NAR DOI: 10.1093/nar/gkn061). In agreement with these studies we observed co-precipitation of U6 snRNA with Larp7 at lower efficiency compared to 7SK and β -actin mRNA. Interestingly, we obtained an analogous result, i.e. co-purification of high levels of β -actin mRNA and lower levels of U6, when we performed RNA immunoprecipitation with Mepce. These data suggest that β -actin mRNA is associated with 7SK particles containing Mepce and Larp7 and might contribute to the association of 7SK with Smn and hnRNP R. We inserted the RNA immunoprecipitation data as Supplementary Fig. 5e and f.

It is unclear why LARP7 MEPCE 7SK complex would interfere with the assembly of some snRNP (U2) but not others (U1)? This should be discussed more in detail.

Author's response: The reviewer raises an important question about the differential sensitivity of snRNPs towards changes in SMN complex activity. Importantly, the snRNAs that we found to be perturbed most by transcriptional inhibition (U4atac and U11 snRNA levels, U2 and U2 snRNP assembly) are the same that are affected in spinal cords of SMA mouse models expressing low levels of SMN (Gabanella et al. 2007 Plos One DOI: 10.1371/journal.pone.0000921, Zhang et al. 2008 Cell DOI: 10.1016/j.cell.2008.03.031). The biochemical basis for this differential susceptibility of individual snRNAs to low SMN complex activity is presently unknown but might be due to differences in their structure or their affinity for the SMN complex. We now added a paragraph to the Discussion in which we compare our results with those from studies on SMA models and in which we speculated on the reasons for the differential sensitivity of some snRNAs to changes in Smn complex activity due to 7SK binding:

"We found that transcriptional inhibition particularly affected the snRNA levels of U4atac and U11, and the assembly of U2 and U12 snRNP particles. This suggests that these snRNAs are more sensitive towards changes in transcription rates or towards alterations in Smn complex activity following association with the 7SK complex. Interestingly, U2, U4atac, U11 and U12 are among those snRNPs that are most strongly reduced in the spinal cord of SMA mouse models harboring low levels of Smn^{42,43}. To our knowledge, the biochemical basis for the differential susceptibility of individual snRNAs to reduced SMN complex activity is presently unknown. It is possible that snRNAs differ in their affinity for the SMN complex or in their efficiency of Sm core assembly due to structural differences. Either way, the reduction of particular snRNPs that occurs upon prolonged inhibition of snRNP assembly following SMN depletion might induce splicing defects of introns that are particularly dependent on them such as U12-dependent introns⁴⁴."

42. Zhang, Z. et al. SMN deficiency causes tissue-specific perturbations in the repertoire of snRNAs and widespread defects in splicing. Cell 133, 585-600 (2008).

43. Gabanella, F. et al. Ribonucleoprotein assembly defects correlate with spinal muscular atrophy severity and preferentially affect a subset of spliceosomal snRNPs. *PLoS. One.* 2, e921 (2007).

44. Doktor, T. K. et al. RNA-sequencing of a mouse-model of spinal muscular atrophy reveals tissue-wide changes in splicing of U12-dependent introns. *Nucleic Acids Res.* 45, 395-416 (2017).

Reviewer #3 (Remarks to the Author):

In the present article entitled “Transcription regulates snRNP production through interaction of 7SK with the Smn complex” Changhe Ji et al. investigate the association of Smn complex with 7SK complex involved in transcriptional regulation providing important information about how transcriptional control is linked to the spliceosome and biogenesis. The manuscript is well done, writing is concise and simple and figures are very complete and illustrative. Before its publication in Nature Communications I would suggest doing these little changes:

Figures:

Fig 1a. Maybe it could be illustrative to highlight in the volcano plot the protein Larp7 too as authors mention it in the text.

Author’s response: We have included Larp7 in the volcano plot in Fig. 1a.

Discussion:

Line 459. Remove the question.

Author’s response: We have removed the question.

Materials and Methods:

Line 660. Clarify briefly the reason to use LysC for digestion prior trypsin.

Author’s response: Initial digestion with LysC was conducted to facilitate cleavage of sites that might not be accessible for trypsin. LysC is stable in 8M urea, thus it can be used for initial digestion at very denaturing conditions. These strongly denaturing conditions enhance digestion at otherwise poorly accessible cleavage sites, for instance if the cleavage sites are in stable protein structures or hydrophobic regions that might aggregate. This initial digestion ameliorates the accessibility problem and generates more soluble peptides which can subsequently be cleaved by trypsin. While digestion of peptides by trypsin is optimal for proteomic analysis due to the length, ionizing and fragmenting properties of the resulting tryptic peptides, trypsin is not stable at 8M urea and would be denatured by such a high urea concentration. Dilution of the sample reduces the urea concentration and permits digestion by trypsin. This is a common approach in proteomics and has been used before, for instance here:

Hornburg D, Drepper C, Butter F, Meissner F, Sendtner M, Mann M. Deep proteomic evaluation of primary and cell line motoneuron disease models delineates major differences in neuronal characteristics. *Mol Cell Proteomics*. 2014 Dec;13(12):3410-20. doi: 10.1074/mcp.M113.037291.

To better explain the use of LysC we modified the relevant section as following:

“Proteins were denatured by adding 150 µl of 8 M urea 50 mM Tris-HCl pH 7.5 solution and 5 µl of 30 µM dithiothreitol and then digested by LysC (0.25 µg/sample) at RT. This initial protein digestion with LysC under strongly denaturing conditions facilitates cleavage at otherwise poorly accessible sites. To permit subsequent digestion with trypsin, which is not stable at 8 M urea, samples were diluted 4-fold with 50 mM Tris-HCl pH 7.5.”

Data in PRIDE: is the data already published in PRIDE? Otherwise it is good to provide username and password for reviewers in the manuscript.

Author's response: We have deposited the data in PRIDE under the dataset identifier PXD021360. The reviewer account details are: Username: reviewer_pxd021360@ebi.ac.uk, Password: MkPPNZ8F. We have included the login details in the cover letter but not in the manuscript. We apologize for this.

REVIEWERS' COMMENTS

Reviewer #1 (Remarks to the Author):

In the revised version, the authors have addressed the critiques to my satisfaction. And they did so with remarkable speed!

The changes they made strengthen the manuscript and tie together a number of loose ends from the previous literature about 7SK, hnRNP proteins and SMN. Congratulations on a fine study.

Signed,
Greg Matera

Reviewer #2 (Remarks to the Author):

In this revised version and in the included response to reviewers document the authors have answered in detail to the comments of all three reviewers. The manuscript should be published without further modifications.

Reviewer #3 (Remarks to the Author):

The authors have adequately responded to all my concerns. The manuscript is now ready for its publication in Nature Communications.